# BrainPhys neuronal medium optimized for imaging and optogenetics in vitro

Michael Zabolocki[1,2,9], Kasandra McCormack[3,9], Mark van den Hurk [1], Bridget Milky[1,2],
Andrew P. Shoubridge[1], Robert Adams [1,2], Jenne Tran[1,2], Anita Mahadevan-Jansen[4], Philipp Reineck [5],
Jacob Thomas[6,7], Mark R. Hutchinson [6,7], Carmen K. H. Mak [3], Adam Añonuevo[3], Leon H. Chew[3],
Adam J. Hirst[3], Vivian M. Lee[3,6,8], Erin Knock[3] & Cedric Bardy [1,2✉]

The capabilities of imaging technologies, fluorescent sensors, and optogenetics tools for cell biology are advancing. In parallel, cellular reprogramming and organoid engineering are expanding the use of human neuronal models in vitro. This creates an increasing need for tissue culture conditions better adapted to live-cell imaging. Here, we identify multiple caveats of traditional media when used for live imaging and functional assays on neuronal cultures (i.e., suboptimal fluorescence signals, phototoxicity, and unphysiological neuronal activity). To overcome these issues, we develop a neuromedium called BrainPhys™ Imaging (BPI) in which we optimize the concentrations of fluorescent and phototoxic compounds. BPI is based on the formulation of the original BrainPhys medium. We benchmark available neuronal media and show that BPI enhances fluorescence signals, reduces phototoxicity and optimally supports the electrical and synaptic activity of neurons in culture. We also show the superior capacity of BPI for optogenetics and calcium imaging of human neurons. Altogether, our study shows that BPI improves the quality of a wide range of fluorescence imaging applications with live neurons in vitro while supporting optimal neuronal viability and function.

[1] Laboratory for Human Neurophysiology and Genetics, South Australian Health and Medical Research Institute (SAHMRI), Adelaide, SA, Australia. [2] College of Medicine and Public Health, Flinders University, Adelaide, SA, Australia. [3] STEMCELL Technologies, Vancouver, Canada. [4] Vanderbilt Biophotonics Centre, Vanderbilt University, Nashville, USA. [5] ARC Centre of Excellence for Nanoscale BioPhotonics & School of Science, RMIT University, Melbourne, VIC, Australia. [6] Adelaide Medical School, University of Adelaide, Adelaide, SA, Australia. [7] Australian Research Council Centre of Excellence for Nanoscale BioPhotonics, University of Adelaide, Adelaide, SA, Australia. [8] Universal Cells, Seattle, USA. [9] These authors contributed equally: Michael Zabolocki, Kasandra McCormack. ✉email: cedric.bardy@sahmri.com

For many years, light microscopy was restricted to anatomical and cellular morphology analysis. In the last two decades, technological advances in fluorescent probes, optogenetics, camera sensors, light-emitting diodes, and microscopes have unleashed the capacity of light microscopy for functional imaging[1–7]. For example, calcium imaging is an increasingly popular electrophysiological assay[8] and neuroscientists are slowly replacing laborious patch-clamping experiments with optical voltage sensors[6,9,10]. Precise light-emitting molecular pH sensors provide insights into synaptic or lysosomal mechanisms. At the same time, light has also become a powerful tool to remotely control the function of neurons. Optogenetics allow precise activation and inhibition of subtypes of neurons[7,11,12]. The design of photoreleasable caged molecules advances our understanding of molecular physiology[13–15]. However, unintended light-induced cytotoxicity with live-cell imaging technologies in vitro may hinder scientific discoveries[16]. Limiting the intensity and duration of light exposure or using longer wavelengths (>500 nm) may partially reduce live imaging-induced cytotoxicity[17–20]. However, it does not resolve the issue and comes at the cost of sub-optimal signal-to-background ratios.

Neuronal cultures are notoriously difficult to maintain and have a low tolerance for manipulation and stress. The most likely cause of light-induced toxicity is the generation of reactive oxygen species produced by light-reactive components in the media[21–23]. Previous studies have shown that adjusting phosphate buffers such as N-(2-hydroxyethyl)piperazine-N′-2-ethane sulfonic acid (HEPES), may reduce the accumulation of light-induced cytotoxic products in tissue culture media[24]; further removing riboflavin from Neurobasal™-based and DMEM-based media improved rat primary neuronal cell survival when exposed to blue light (470 nm)[25]. These base media optimized for phototoxicity may help with some imaging experiments. However, we found that with human neuronal models, DMEM and neurobasal are suboptimal for the firing of action potentials and synaptic activity (glutamatergic and GABAergic), which is a major caveat to modelling the physiology of the human brain in vitro[26].

Thus, there is a need for a neuronal medium specialized for functional imaging experiments. Such a medium needs to support neurophysiological activity, minimize light-induced toxicity, and enhance fluorescent signal-to-background ratios. To address this need, we implemented the formulation of the original BrainPhys™ (BP) medium[26] in the design of a functional imaging neuromedium, which we called BrainPhys™ Imaging (BPI). We benchmarked its performance in comparison with other media currently available. We found that removing the light-reactive components from BP enhanced fluorescent signals and reduced neuronal phototoxicity. We demonstrated that the BPI formula could be used to culture neurons without a detrimental effect on cell survival, and its osmolality matches physiological levels of human CSF. Finally, we used patch-clamping, multielectrode arrays, calcium imaging, and optogenetics to show that the formulation of BrainPhys™ Imaging supports optimal electrophysiological functions of human neurons in vitro.

## Results

**Adjustment of standard BrainPhys basal medium formula to optimize live-cell imaging.** To optimize the original BrainPhys basal medium for imaging, we adjusted the components within the formulation which we identified as potentially increasing fluorescent background noise and phototoxicity. The major changes made were the removal of phenol red, and adjustments of the vitamins (e.g., riboflavin) and pH buffers. The vitamin content of the medium appeared as the main factor responsible for phototoxicity and autofluorescence interference at excitation

wavelengths less than 500 nm. The adjustments made had no significant effect on the overall osmolality of the medium (Fig. 1a). This is an essential attribute of BrainPhys that we wished to maintain as it is similar to that of human cerebrospinal fluid (~300 mOsmol/L). We also confirmed that the osmolality of media conditioned with hiPSC-derived neurons was identical to the fresh media and stable over time (tested for up to 2 weeks, Supplementary Fig. 1a). The pH was also maintained at physiological levels (pH 7.4) in the BPI formulation. The formulation is referred to as BrainPhys Imaging (BPI). For comparison, we also evaluated two other commercially available imaging media, BrightCell™ NEUMO and FluoroBrite™ DMEM, which were previously modified based on Neurobasal and DMEM, respectively[25]. We previously demonstrated that the more physiological composition of BrainPhys basal medium improves the electrophysiology and synaptic activity of human neurons in vitro in comparison to DMEM and Neurobasal[26]. In the present study, we compared the imaging performance, phototoxicity, and electrophysiological properties of cells in BPI to standard neuronal media and other imaging media currently available.

**BPI minimizes absorbance and autofluorescence across the visible light spectrum.** We compared the performance for fluorescence imaging of BPI to other tissue culture media (Figs. 1 and Fig. 2). First, we measured the absorbance spectrum (300–800 nm) of BPI basal and other basal media without supplements (Fig. 1b). The absorbance above 600 nm was close to zero for all media tested. However, below 600 nm the relative level of absorbance differed between basal media. All classical basal media used for neuronal culture (standard BrainPhys, Neurobasal, and DMEM/F12) had relatively high absorbance for all wavelengths below 600 nm. Removing phenol red from standard media helped to reduce the absorbance between 400 and 600 nm. Further adjustment of vitamins and pH buffer in BPI also helped to reduce the absorbance below 400 nm to levels almost as low as PBS. We also confirmed that the addition to the basal medium of a range of supplements (SM1, N2-A, BDNF, GDNF, Ascorbic Acid, cAMP, Laminin; selection based on the standard BrainPhys formulation established in ref. [26]) did not increase the absorbance above 300 nm (Fig. 1b).

Then, we measured the relative autofluorescence of each media in the absence of cells (Fig. 1c–f). We used a custom-made bifurcated fiber probe that we positioned within the media. We then excited each medium at four wavelengths sequentially from violet to green (375, 405, 488, and 532 nm) and measured the emission spectra from 400 to 700 nm. We found that BPI basal medium autofluorescence across the visible light spectrum was as low as for PBS (Fig. 1c). In contrast, all other standard media tested (BP, DMEM, Neurobasal) showed much higher autofluorescence signals at excitation wavelengths from violet to green. When tested with short wavelengths light excitation (375 and 405 nm), BPI also showed slightly lower autofluorescence than NEUMO and FluoroBrite (Fig. 1c). Low autofluorescence of BPI across the light spectrum was maintained despite adding supplements. For further statistical comparison, we repeated these experiments using a plate reader (Fig. 1d–f). We compared the mean fluorescence intensity with three dichroic filters: blue (excitation/emission at 355/460 nm), green (excitation/emission at 485/520 nm), and red (excitation/emission at 544/590 nm). The blue, green, and red autofluorescence levels of BPI were significantly lower than that of any other standard basal neuromedia (BrainPhys, DMEM, Neurobasal) (Fig. 1d–f). At shorter wavelengths (blue and green), BPI also outperformed the autofluorescence of other specialized imaging media (BrightCell™ NEUMO and FluoroBrite™ DMEM) (Fig. 1d, e). At longer

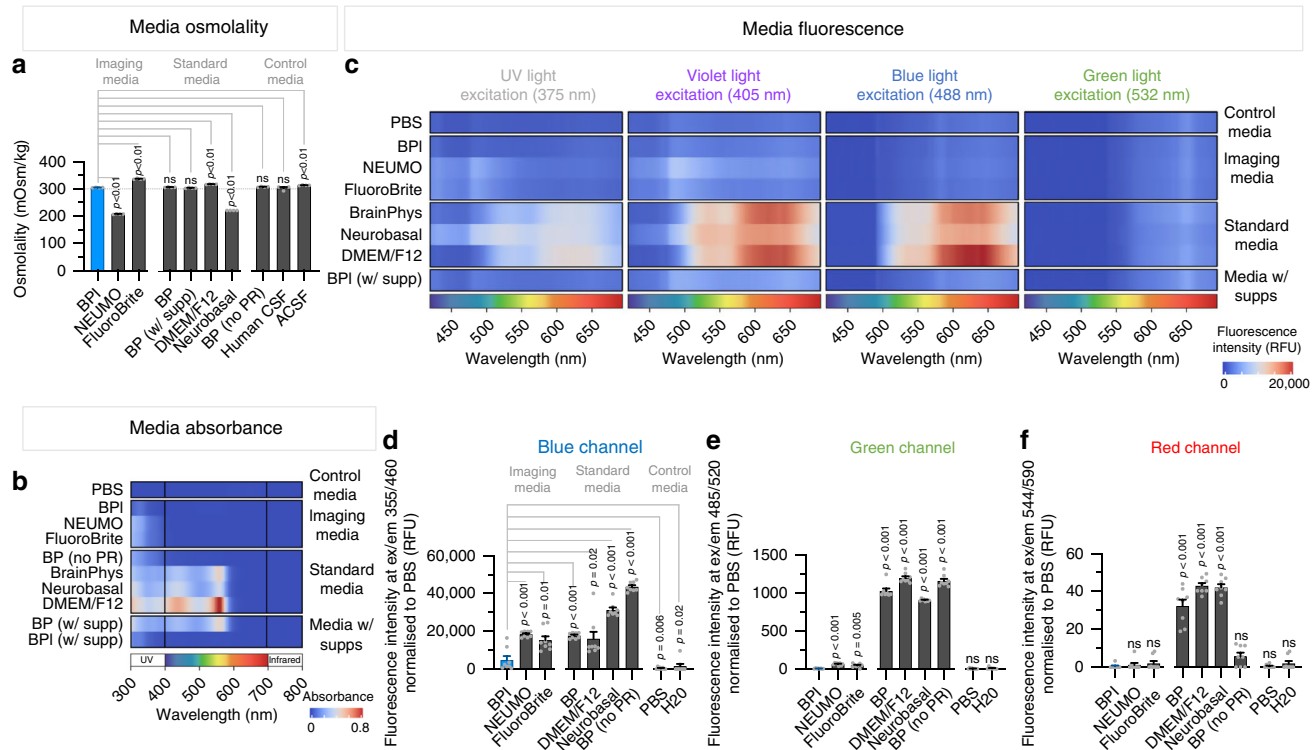

**Fig. 1 BrainPhys Imaging reduces auto-fluorescence and light absorbance. a** The osmolality of BrainPhys Imaging (BPI) is similar to human cerebrospinal fluid (hCSF) (~305 mOsmol/kg). Neurobasal and NEUMO were significantly lower (210–220 mOsmol/kg) and DMEM/F12 and FluoroBrite higher (317–338 mOsmol/kg). Media osmolality data was collected from four replicates per condition. A total of three hCSF samples, each pooled from up to four subjects, were tested across three independent experiments ($n = 3$). See also Supplementary Fig. 1a. **b–f** Comparison of the optical properties of BPI with other basal media specialized for imaging (NEUMO, FluoroBrite), standard basal neuromedia (BrainPhys, BP; BrainPhys without phenol red, BP no PR; DMEM/F12; Neurobasal) and control media (phosphate-buffered saline, PBS; deionized water, $H_2O$). **b** The absorbance spectra from 300 to 800 nm acquired from basal media alone (without cells), and after adding supplements required for the long-term culture of brain cells. Virtually all fluorophores used for cell imaging require light stimulation above 300 nm. **c–f** The mean autofluorescence intensities of basal media (without cells) across the entire visible spectrum. BPI shows autofluorescence intensities similar to PBS. **c** The emission spectra across 400–700 nm captured for the 375 (ultra-violet), 405 (violet), 488 (blue), and 532 nm (green) excitation wavelengths from test and control media. **d–f** Autofluorescence at 460, 520, and 590 nm emission wavelengths were measured following excitation at 355, 485, and 544 nm, respectively. Results were generated from eight replicate wells per medium ($n = 8$) analyzed across three independent experiments. For normalization, the mean fluorescence intensity in PBS was subtracted from the other media. Data are presented as mean ± SEM. Significance determined via two-tailed nonparametric unpaired (Mann–Whitney) tests. ns, $P > 0.05$.

wavelengths (red), the autofluorescence of BPI and other imaging media was similar (Fig. 1f). The green and red autofluorescence of BPI was equivalent to that of PBS and $H_2O$ (Fig. 1e, f). However, we noted that the blue autofluorescence in BPI remained slightly higher than PBS or $H_2O$, despite outperforming all the other media (Fig. 1d). The removal of phenol red alone did not show any reduction in autofluorescence in the blue and green channels, but helped to lower the red autofluorescence (Fig. 1d–f). Autofluorescence with light excitation at a longer wavelength (532 nm) was relatively low for all media tested (Fig. 1c), though the plate reader experiments revealed a relatively small but significant reduction in autofluorescence for BPI compared to standard media in the red channel (Fig. 1f). Overall, the absorbance and autofluorescence of BPI were most distinctively reduced for excitation wavelengths between ultraviolet and blue compared to other neuronal media (Fig. 1b, c). Most importantly, the modifications made in BPI lowered the absorbance and autofluorescence at levels similar to PBS across all visible wavelengths.

**BPI improves fluorescent signal-to-background ratios**. We then showed that the lower level of autofluorescence of BPI enhanced signal-to-background ratios, when imaging brain cells in vitro,

using healthy human iPSC-derived neurons in monolayers (Fig. 2a–c and Supplementary Figs. 2, 3) and organoids (Fig. 2d) or rat primary cortical neurons (Fig. 2e–f and Supplementary Fig. 1c–f). Live human iPSC-derived neurons were infected with lentivectors to express green (eGFP) using a synapsin promoter (Fig. 2a–c and Supplementary Figs. 2, 3). The human neurons were imaged live in standard BrainPhys or BPI with identical imaging parameters. BPI significantly increased signal-to-background ratios at the neurite and soma regions, and significantly reduced mean background intensities (Fig. 2a–c). When combining analysis at soma and neurite regions, similar increases in signal-to-background ratios were observed. Reperfusing the cells with artificial cerebrospinal fluid (ACSF), after testing them subsequently in BPI and BP, recovered signal-to-background ratios in the background, soma and/or neurite regions to levels identical to BPI (Fig. 2a–c). This confirmed that the effect was not simply due to possible photobleaching. Overall, all the neuronal monolayer and organoid images in BPI showed increased signal-to-background ratios for lower wavelengths (blue and green) compared to all other media tested (standard BrainPhys, Figs. 2a–c, e, f and Supplementary Figs. 1c–f, 2, 3; organoid medium, Fig. 2d; BrainPhys without Phenol Red and NEUMO, Supplementary Fig. 1c–f). The live imaging of an assembly of brain-region specific organoids (assembloids[27]) in BPI also

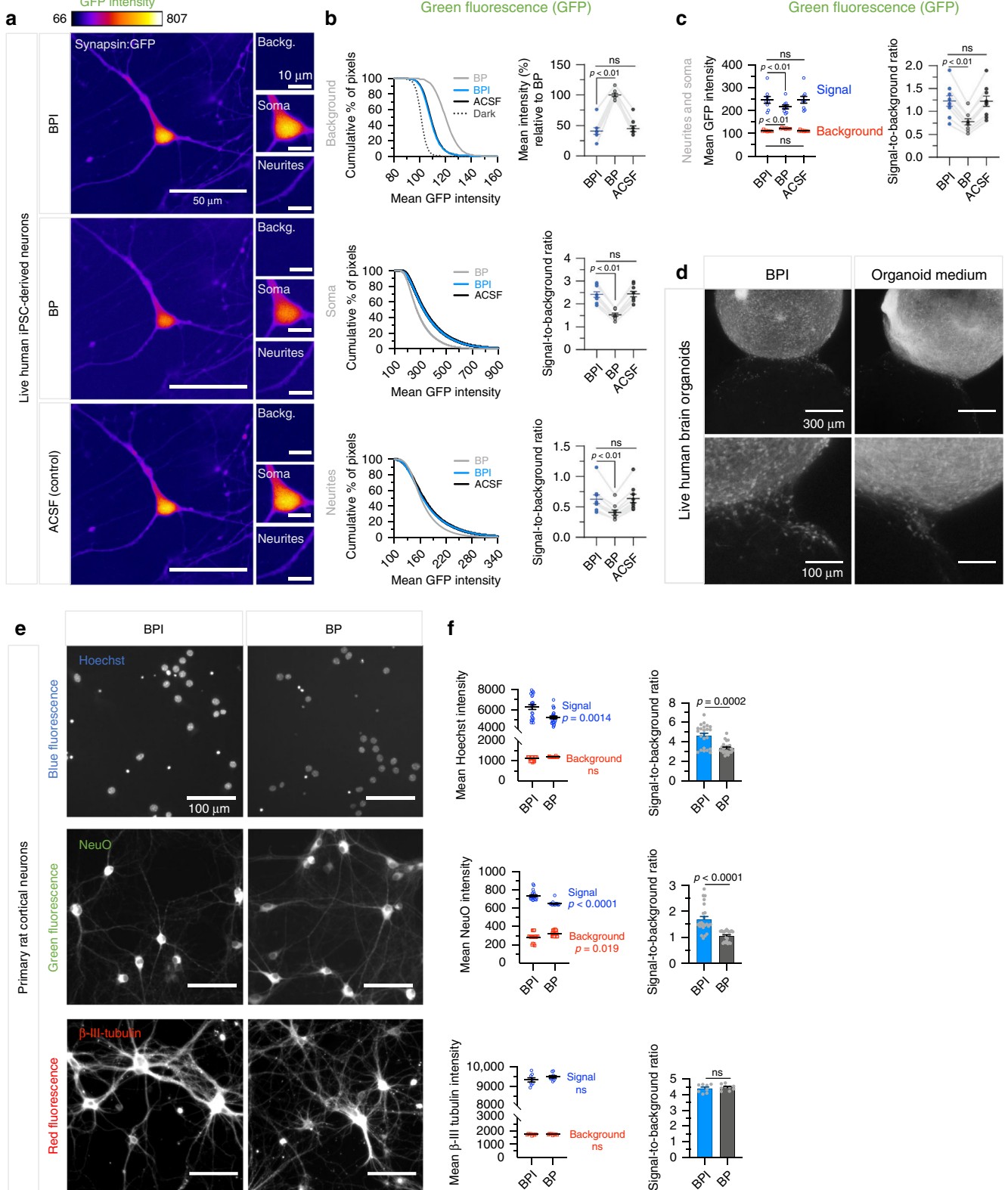

allowed for better visualization of migrating neurons than in standard organoid maturation medium (Fig. 2d). We then used rat cortical primary neurons to test the signal-to-background ratio at different wavelengths. The primary neurons were either fixed and labeled in red with β-III-tubulin and DyLight™ 594 (excitation/emission at 593/618 nm) or live in green with the neuron-specific dye NeuroFluor™ NeuO (excitation/emission at

468/557 nm). For Hoechst™ 33342 (excitation/emission at 361/497 nm), the neurons were either labeled live or fixed first. For each fluorophore, all images were conducted with identical imaging parameters but in different media (Fig. 2e, f and Supplementary Fig. 1c–f). For longer wavelengths (red), we found no significant improvement in signal-to-background ratio with images taken in BPI (Fig. 2e, f and Supplementary Fig. 1e, f).

**Fig. 2 BrainPhys Imaging increases the fluorescence signal-to-background ratio of neuronal cultures.** Fluorescent imaging in BrainPhys imaging (BPI) medium improves signal-to-background ratios. **a–c** Live human PSC-derived neurons expressing GFP and imaged in BPI, BrainPhys (BP) or artificial cerebrospinal fluid (ACSF) display higher mean GFP intensities at soma and neurite regions in BPI or ACSF relative to BP, with reduced background intensities. Data were analysed from the same cells (regions of interest, ROIs) in BP, BPI, or ACSF across nine field-of-views (FOVs) per condition; $n = 293$ ROIs analyzed at the soma, $n = 469$ ROIs analyzed at dendrites, $n = 139$ ROIs analyzed in background. **b** (left) Cumulative percentage (%) of pixels at background, soma and neurite regions in BP (gray), BPI (blue), and ACSF (black). Camera dark counts (without medium and cells) are labeled "dark". **b** (right) Mean intensity values at background regions (top) in ACSF and BPI were significantly reduced relative to BP. Signal-to-background ratios at isolated neurite and soma regions were significantly higher in BPI and ACSF than BP. **c** Analysis of combined neurite and soma regions improved signal-to-background ratios in BPI and ACSF compared to BP. **d** Example images of brain assembloids representing a GFP-labeled ventral organoid fused with a nonlabeled dorsal organoid imaged in Organoid Maturation medium (right) or BPI (left). Increased visibility of interneuron migration is seen in BPI (bottom). **e, f** Representative primary cortical neurons stained and imaged live with Hoechst 33342, NeuroFluor NeuO, and fixed with β-III tubulin-Dylight 594. Images taken in both media are displayed with the following minimum/maximum intensity counts: 0/8500 (Hoechst), 0/3600 (NeuO), 1000/12,000 (β-III tubulin). Signal-to-background ratios were improved when imaging Hoechst in BPI ($n = 23$ FOVs) compared to BP ($n = 24$ FOVs). This was also seen for NeuO imaged in BPI ($n = 23$ FOVs) relative to BP ($n = 22$ FOVs). No significant differences were found when imaging β-III tubulin in either BPI ($n = 8$ FOVs) or BP ($n = 9$ FOVs). Images were collected from two biologically independent experiments (Hoechst/NeuO; β-III tubulin) with one well per condition. Data in **b**, **c**, **f** are presented as mean ± SEM. Significance determined via two-tailed nonparametric paired (Wilcoxon) tests (**b**, **c**) and nonparametric unpaired (Mann–Whitney) test (**f**). Nonsignificant $P$-values >0.05 are annotated with ns.

Overall our data supports that BPI constitutes an alternative imaging neuronal medium that provides lower absorbance, lower autofluorescence, and enhanced fluorescent signal-to-background ratios throughout the entire light spectrum from ultraviolet (300 nm) to infrared (800 nm), which is used by virtually all fluorophores, sensors and imaging probes in cell biology.

**BPI reduces phototoxicity across the light spectrum (from violet to red).** To compare the extent of phototoxicity on neuronal cultures in different media, we first measured the viability of primary rat cortical neurons after prolonged exposure to violet (395–405 nm; 430 Lux ± 3.70 S.E.M), blue (450–475 nm; 14 Lux ± 1.41 S.E.M) and red (620–740 nm; 55 Lux ± 4.99 S.E.M) LED light inside a temperature-controlled incubator (Fig. 3a–d). Neurons exposed to light in phototoxic media exhibited broken neurites and disintegrated cell bodies (Fig. 3a–c and Supplementary Fig. 5); the number of live cells was counted post light-exposure to quantify cell viability. Exposure to violet and blue light caused cell death in standard BrainPhys medium (with or without phenol red) but not in BPI (Fig. 3a–d). We also compared the photo-toxicity of the three wavelengths in BPI to NEUMO. Although NEUMO basal is provided with a supplement (SOS), we were unable to sustain healthy neurons with this combination for long enough to perform this assay (Supplementary Fig. 4b). Thus, we tested NEUMO basal medium combined with the same supplement that was added to BrainPhys basal in these experiments (SM1 Neuronal Supplement). In these conditions, NEUMO performed comparably to BPI when stimulated for 12 h with blue light (Fig. 3d). However, when using higher energy violet light (430 Lux ± 3.70 S.E.M) for 6 h, the cells only survived in BPI (Fig. 3d). Red light did not appear phototoxic in any media we tested, even when exposed for longer at four times the Lux intensity of the blue light test (Fig. 3d). To quantify more subtle phototoxic effect preceding cell death, we exposed BP and BPI media to ambient light in the tissue culture hood for 24 h and fed human PSC-derived midbrain and cortical neurons daily for up to 7 days (half media change), while measuring the accumulation of lactate dehydrogenase (LDH) in the supernatant. LDH is a cytosolic enzyme, which is present in neurons and released into the supernatant upon damage to the plasma membrane. We found that media exposure to ambient light triggered a higher release of LDH in cultures fed with BP compared to BPI within two days (Fig. 3e, f). These results were consistent both with cortical and midbrain human neurons (Fig. 3e, f). Riboflavin, which is present in BP and other classical neuromedia, can release free radicals such as hydrogen peroxide ($H_2O_2$). We found that

$H_2O_2$ levels significantly increased in BP after 24 h exposure to ambient light but not in BPI. This $H_2O_2$ increase was observed despite adding antioxidants in supplements to both media (Fig. 3g). The spectrum of the ambient light from our tissue culture hood showed multiple-wavelength peaks between 400 and 650 nm (Fig. 3h). Therefore, we used an LED controller instead of ambient light to show that wavelengths around 475 nm (Fig. 3i and Supplementary Fig. 4a) were sufficient to increase the level of $H_2O_2$ in BP medium in a dose-dependent manner (Fig. 3j). At the highest dose tested (~105 mW at 475 nm) the level of $H_2O_2$ doubled in BP and only slightly increased in BPI. Oxidative stress is also often associated with mitochondrial impairments. Indeed, we observed a significant reduction of the mitochondrial potential of neurons exposed to violet light in standard BP (Fig. 3k). This effect appeared to precede neuronal death (Supplementary Fig. 4c) and was comparable to exposing the neurons to 100 μM of $H_2O_2$. In contrast, the mitochondrial potential of neurons was not affected by light stimulation in BPI (Fig. 3k). Altogether, BPI was the only neuronal medium tested that showed no neuronal phototoxicity across the full visible light spectrum.

**BPI supports optimal action potentials and synaptic activity in human neurons.** We then asked whether BPI maintains the fundamental electrophysiological properties of human neurons. We used patch-clamping and MEA to test the effect of BPI on the electrophysiological activity of human neurons (Figs. 4 and 5). The neurons were matured from human PSC-derived neural progenitors in standard BrainPhys for 4–5 months before patch-clamping (Supplementary Fig. 6a). The cells were selected for patch-clamping based on their neuronal morphology visualized with synapsin:GFP lentiviral marker (Supplementary Fig. 6b). During the whole-cell recordings, we alternated the perfusion between BPI and artificial cerebrospinal fluid (ACSF), which is used as the gold-standard for acute electrophysiology. We first found that acute perfusion of BPI had no negative effect on the types of action potential (AP) firing patterns, which were defined previously in[28] (AP Types 1–5; Supplementary Fig. 6c). To confirm that BPI does not have more subtle effects, we further examined the electrophysiological properties of "mature" neurons (Type 5). Our results show that the cellular mechanisms critical to generate action potentials were healthy in BPI, and we found no difference in the electrophysiology of the cells when patched in ACSF (Fig. 4 and Supplementary Fig. 6) or in standard BP (Supplementary Fig. 7). To measure evoked action potentials (in current-clamp) or voltage-gated sodium and potassium currents (in voltage-clamp), we maintained the resting potential at

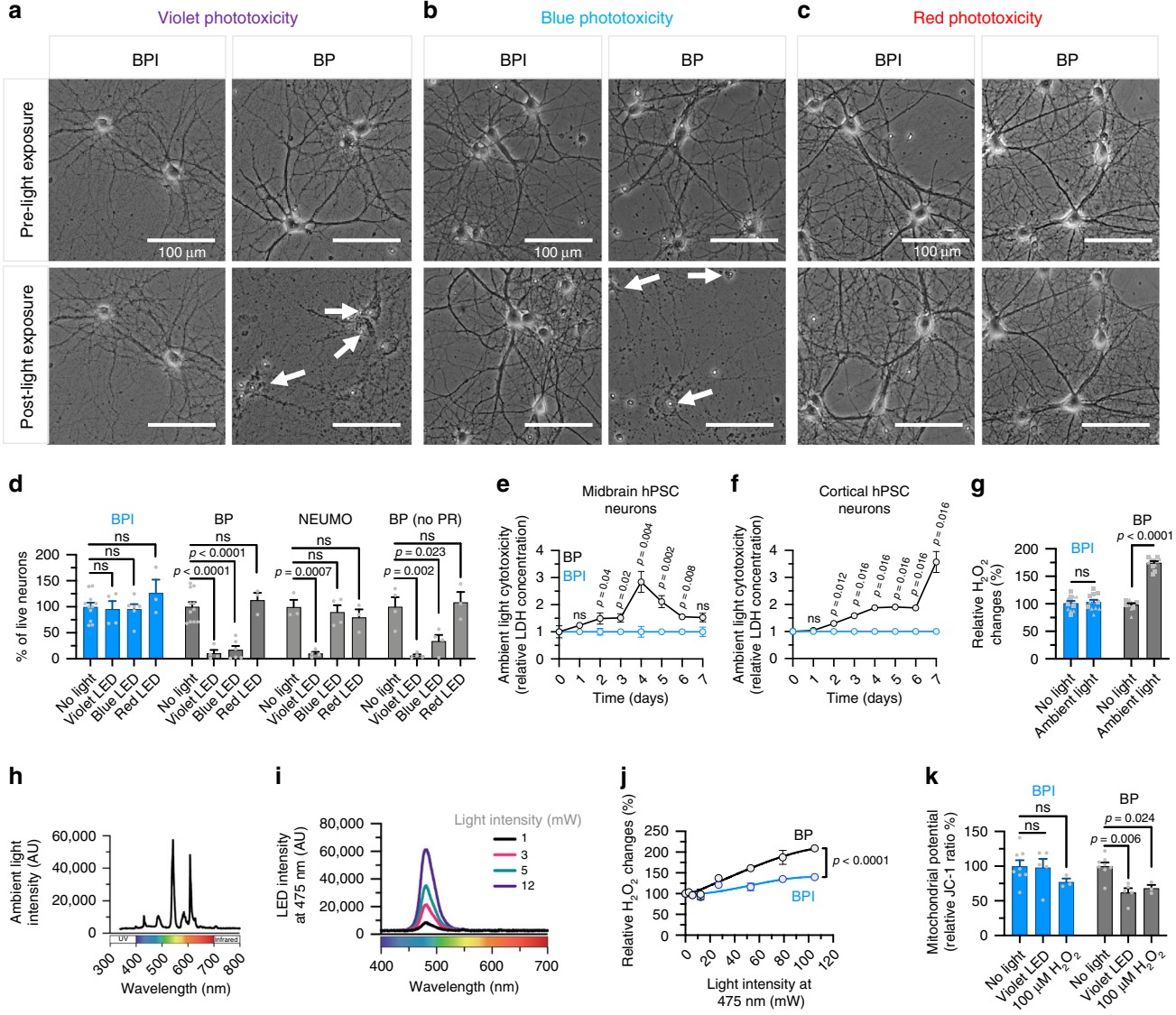

**Fig. 3 BrainPhys Imaging minimizes phototoxicity. a–d** Representative images of rat cortical primary neurons cultured in BrainPhys (BP) and BrainPhys Imaging (BPI) pre and post violet, blue and red LED exposure show dying neurons in BP (highlighted by white arrows). Each image is a representative cropped field-of-view (FOV). See also Supplementary Fig. 5. **d** Only BPI maintained neuronal viability post violet and blue LED exposure compared to BP, NEUMO, and BP without phenol red (BP no PR). Red LED exposure caused no apparent phototoxicity. Data were collected across 3–5 independent experiments, each including 75 field-of-views analyzed across three replicate wells. The number of neurons (per $cm^2$) post-light exposure were normalized to "no light" conditions. **e, f** Significant cytotoxicity in human midbrain (days 0–5: BPI and BP, $n = 6$ wells; days 6–7: BPI and BP, $n = 5$ wells) and cortical neurons (BP, $n = 5$; BPI, $n = 4$ wells) was seen when BPI and BP media (with supplements) were exposed to ambient tissue culture light for 24 h and fed to cells across 7 days. **g** Reactive oxygen species ($H_2O_2$) in BP (without cells) was significantly increased following 24 h stimulation with ambient light, relative to BPI ($n = 12$ wells per condition). Symbols represent cortical (triangle) or midbrain (square) supplements. **h** Ambient tissue culture light emission spectrum. **i** Blue LED light emission spectrum. **j** Relative $H_2O_2$ levels of BPI and BP (without cells) after 24 h stimulation with blue LED light flashes over increasing power intensities ($n = 6$ wells per condition). Data shown were normalized to "no light" conditions. See also Supplementary Fig. 4a, c. **k** Mitochondrial health of primary rat cortical neurons in BPI and BP (supplemented with SM1) was assessed using a mitochondrial membrane potential indicator (JC-1). Following 1 h violet LED light exposure or 100 µM $H_2O_2$ treatment, only neurons exposed to BPI maintained mitochondrial health. Data were collected across 3–5 independent experiments, across three biological replicates. Results are shown normalized to "no light'" conditions. See also Supplementary Fig. 4c. Data in **d–g**, **j–k** are presented as mean ± SEM. Significance determined via two-tailed unpaired $t$-test (**d**), two-tailed nonparametric unpaired (Mann–Whitney) test (**e–g**, **k**) and one-sided sum-of-squares $F$-test (**j**). Nonsignificant $P$-values >0.05 are annotated with ns.

−70 mV (by injecting on average 50 pA of current) and applied incremental depolarizing steps of current/voltage (Fig. 4a and Supplementary Fig. 6d). Voltage-gated sodium/potassium currents, rheobase, AP amplitude, AP frequencies, and AP hyperpolarizing amplitudes were similar when the recording was made in BPI or in ACSF (Fig. 4c–i). When no current was injected in the patch-clamped neurons, their resting membrane potential was around −50 mV and their AP threshold approached physiological levels (−40mV) both in BPI and ACSF (Fig. 4c). No significant differences were found between the membrane resistance and capacitance of neurons patch-clamped in either BPI or ACSF (Fig. 4j, k). Our results demonstrate the capacity of human neurons to generate optimal action potentials in BPI.

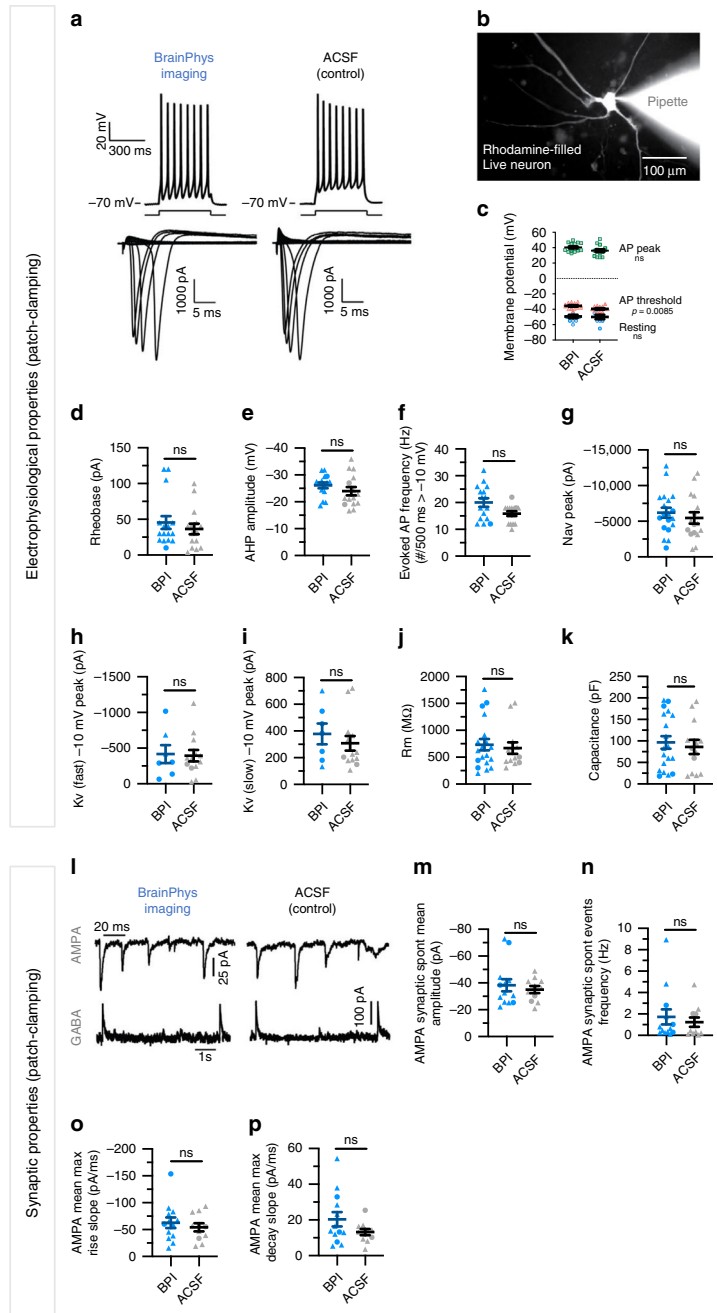

**Fig. 4 BrainPhys Imaging supports optimal electrophysiological and synaptic activity of human neurons in vitro.** Single-cell patch-clamp recordings of iPSC-derived neurons matured in standard BrainPhys (BP)+ supplements medium for >12 weeks, and patch clamped in BrainPhys imaging (BPI) or artificial cerebrospinal fluid (ACSF). All patch-clamped neurons (n = 35) included in the analysis were classified as "Type-5" and patched from a total of 18 coverslips. See also Supplementary Fig. 6. A subset of neurons (n = 6) were recorded in both media. Each point on the graphs represents a single neuron. **a** (Top) Typical evoked action potential (AP) traces following a 500 ms depolarizing current step when patched in ACSF or BPI. **a** (Bottom) Current–voltage characteristics (I–V curve) with +5 mV current steps increments from rest at −70 mV. Voltage-dependent sodium (Nav) and potassium (Kv) current traces shown, respectively, below and above the x-axes. **b** Shows an example image of a single neuron filled with rhodamine following patch-clamp recordings. **c** Pooled data summarizing peak and threshold AP amplitudes (BPI, n = 16; ACSF, n = 15), and resting membrane potential values (BPI, n = 12; ACSF, n = 9). **d–f** Action potential properties (AP) were similar in BPI (n = 16) and ACSF (n = 15). **d** Rheobase values. **e** Peak afterhyperpolarization (AHP) amplitudes. **f** Maximum firing frequency of evoked APs (with amplitudes > −10 mV). **g–i** Current–voltage characteristics in BPI were similar to ACSF perfusate. **g** Peak NaV current amplitudes (BPI, n = 19; ACSF, n = 16). **h, i** Peak amplitudes of rapidly and slowly inactivating Kv currents (BPI, n = 7; ACSF, n = 13). **j, k** Membrane resistance (Rm) and capacitance values were similar between BPI (n = 19) and ACSF (n = 13). **l–p** Synaptic events mediated by AMPA receptors (excitatory postsynaptic currents, ePSCs) and GABAa receptors (inhibitory postsynaptic currents, iPSCs) were supported in both BPI (n = 13) and ACSF (n = 11) perfusates. **l** Typical spontaneous ePSC (top) and iPSC traces (bottom). **m–p** Average properties of spontaneous synaptic events recorded for 4 min. **m** Mean ePSC amplitudes. **n** Mean ePSC event frequency. **o** Max rise slopes of ePSCs. **p** Max decay slopes of ePSCs. Symbols in **d–k, m–p** represent human neurons tested first (triangles) or second (circle) in either medium. Data are presented as mean ± SEM. Significance determined via two-tailed nonparametric unpaired (Mann–Whitney) tests. ns, P > 0.05.

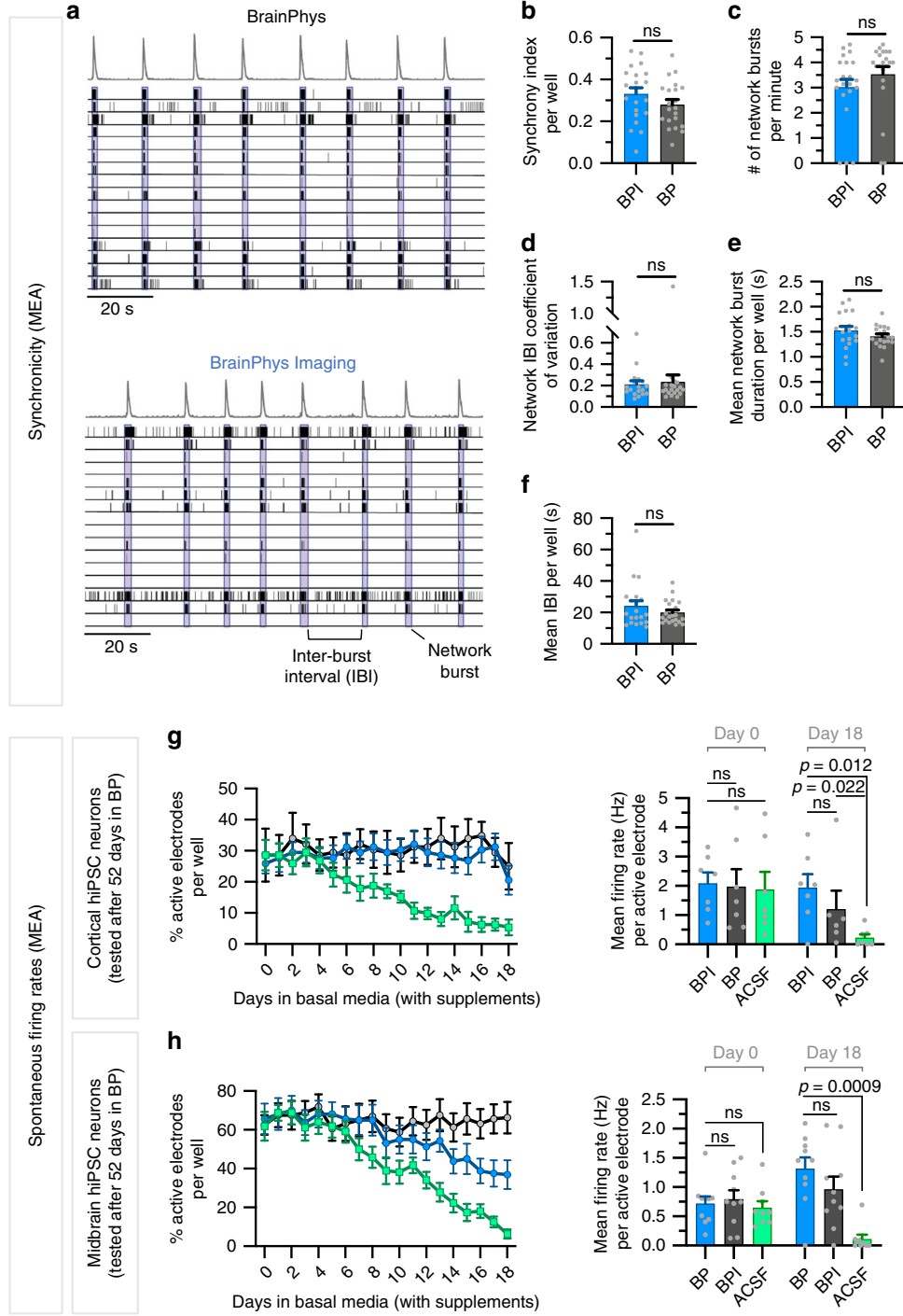

**Fig. 5 BrainPhys Imaging supports neural network synchrony and long-term electrophysiological activity of human neuronal populations in vitro.**
**a–f** Midbrain human iPSC-derived neurons were matured for 100 days in BP before switching to standard BrainPhys (BP) or BrainPhys Imaging (BPI) for 2 h. The same midbrain supplements were added to all basal media. Network activity was compared on 48-well multielectrode array (MEA) plates over 7-min time intervals. Each value represents the average recorded across 16 electrodes per well, with a total of 18–21 wells per condition. **a** Example 2-min raster plots of neuronal network activity in BPI or BP show similar network activity (blue boxes) and spike histograms (gray, top). **b–f** Quantification of MEA network activity showed no significant differences between BPI and BP media for: **b** synchrony index ($n = 21$ wells per condition), **c** number of network events per minute ($n = 21$ wells per condition), **d** network interburst interval (IBI) coefficient of variation ($n = 18$ wells per condition), **e** mean network burst duration (BPI, $n = 18$ wells; BP, $n = 19$ wells) and **f** mean IBI duration (BPI, $n = 20$ wells; BP, $n = 21$ wells). **g**, **h** Cortical and midbrain human iPSC-derived neurons were matured for 52 days in standard BP. Spontaneous firing rates were then compared using MEA recordings over 18 days in BP (black), BPI (blue) and artificial cerebrospinal fluid (ACSF; green). The same cortical or midbrain supplements were added to all three basal media. Spontaneous MEA activity was recorded for 10 min every 24 h across 16 electrodes per well for 29 wells of midbrain (9–10 wells per condition) and 20 wells of cortical (6–7 wells per condition) cultures. For both midbrain and cortical cultures, the spontaneous firing activity was similar in BP, BPI, and ACSF at first but progressively decreased in ACSF after several days. Data in **b–h** are presented as mean ± SEM. Significance determined via two-tailed nonparametric unpaired (Mann–Whitney) tests. Nonsignificant $P$-values >0.05 are annotated with ns.

Neural network communication is achieved via synaptic integration and a regulated balance of inhibition and excitation. In the human brain, the neurotransmitter glutamate is responsible for the majority of synaptic excitation, while GABA is responsible for the majority of synaptic inhibition. The summation of excitatory and inhibitory neurotransmitters governs the probability of postsynaptic action potential firing. Hence, we assessed whether BPI supports basic synaptic activity in human neurons. Voltage-clamp recordings from mature human neurons highlighted spontaneous synaptic events mediated by both AMPA and GABA receptors, when patched in BPI medium or ACSF (Fig. 4l). Voltage clamping at the reversal potential of anions ($-70$ mV) and cations (0 mV) was used to distinguish between glutamatergic ($Na^+$ current-mediated) and GABAergic ($Cl^-$ current-mediated) events. We found no significant differences in any of the properties of AMPA-mediated synaptic events (amplitude, frequency, and kinetics) when mature human neurons were patch-clamped in BPI or in ACSF (Fig. 4m–p). The functional quality of the neuronal network can also be assessed by measuring the synchronicity of action potential firing on multielectrode arrays. We found similar synchronicity properties when comparing the activity of human neurons side-by-side either in BPI or BP with the same supplements (Fig. 5a–f). Altogether, our results demonstrate the capacity of human neurons to generate and integrate optimal synaptic communication in BPI.

The electrophysiology of the cells in standard BrainPhys or BPI media is equivalent to gold standard ACSF. However, maintaining neurons in ACSF for more than a few hours can induce signs of neurodegeneration. We also previously showed that ACSF is not adapted to culture neurons for extended periods of time even when adding supplements[26]. Here, we repeated these results by comparing the function of human cortical and midbrain neurons on MEA over 18 days either in standard BP basal, BPI basal, or ACSF (Fig. 5g, h). In these experiments, we added to all three basal media the same sets of supplements (N2-A, BDNF, GDNF, vitamin C, cAMP, laminin, and either SM1 for midbrain or IGF-1 and SM1 without vitamin A for cortical). The spontaneous electrical activity was similar at first, but after a few days, we observed a significant decline when cultured in ACSF. In contrast, over the same period, the firing activity of cortical and midbrain neurons remained strong in standard BrainPhys medium and BPI (Fig. 5g, h).

**Functional live-cell imaging of human neuronal culture in BPI medium**. Specific ions such as calcium ($Ca^{2+}$) play essential roles in electrophysiological activity and vesicular release at synaptic terminals, and are commonly used as a proxy to measure electrophysiological activity with functional imaging. The level of $Ca^{2+}$ in BP and BPI matches human brain physiological levels (1.1 mM $CaCl_2$). We tested the potential of BPI to image the spontaneous calcium activity of live human neurons. We found that functional imaging can be performed in BPI with optimal performance equivalent to neurophysiological salt solution ACSF, and improved performance compared to NEUMO and Fluoro-Brite imaging media (Fig. 6). For signal quantification, we monitored changes in intracellular $Ca^{2+}$ with a calcium-sensitive dye (Fluo-4 AM) while perfusing different media. Thousand two hundred and twenty-five regions of interest (ROIs) were selected at the soma of neurons across a total of thirteen field-of-views (FOVs) (Fig. 6a). The FOVs were imaged over 4 min at 5 Hz in each of the following perfusates: ACSF, BPI, FluoroBrite, and NEUMO media. To investigate the quality and kinetics of spontaneous calcium activity, we categorized the cells based on the type of detected events into (i) cells with calcium spikes (fast-rising phase events), (ii) cells with calcium waves (slow rising phases), (iii) cells with calcium waves & spikes (combined), or (iv) cells without any spontaneous activity (Supplementary Fig. 8a). BPI did not affect the frequency and amplitude of unitary events (fast-rising phase events, d$F/F > 5\%$) in comparison to ACSF (Fig. 6b–d and Supplementary Fig. 8, Supplementary Movie 1). A similar proportion of fast-rising cell types was measured, when functional imaging was performed in ACSF compared to BPI (Fig. 6e, f and Supplementary Fig. 8). However, a significant decline in the number, proportion, and frequency of fast-rising cell types was observed, when the perfusates were switched to NEUMO or FluoroBrite media (Fig. 6i, j and Supplementary Fig. 9). For slow rising calcium events, no significant changes in kinetics and cell event types were identified when functional imaging was conducted in BPI compared to ACSF, NEUMO, or FluoroBrite (Fig. 6g, h, k, l and Supplementary Figs. 8, 9). The activity of the majority of these cell types was silenced with a voltage-gated sodium channel antagonist (tetrodotoxin, TTX; 1 µM), demonstrating that the calcium events observed are largely mediated by spontaneous action potential activity (Fig. 6e–g, i, k and Supplementary Figs. 8, 9b–e). We also alternated randomly the sequence of media tested to eliminate the possibility of order bias. Altogether, our results indicate that BPI provides optimal conditions for functional imaging of live human neurons in vitro.

**BPI for optogenetics of human neurons**. A panoply of optogenetic tools enables remote control of neuronal function with light[2,7]. To test that BPI is suitable for optogenetics experiments, we transfected human neurons in vitro with a lentiviral vector, to drive the expression of channel-rhodopsin tagged with a yellow fluorescent protein (ChETA-eYFP, which is based on ChR2 with two amino acid modifications to improve the kinetics and amplitude of light-evoked currents[11]). ChETA-eYFP was placed after a human synapsin promoter region (hSyn) to target the expression into neuronal cells. We patch-clamped human neurons expressing ChETA-eYFP (Fig. 7a). Flashes of blue light (5 ms, 100 ms intervals, at 475 nm) evoked action potentials in BPI comparable to ACSF in all neurons (Fig. 7b). Using the same light intensity and duration, we found that ten flashes of light at 10 Hz evoked action potentials at a success rate of 99–100% both in ACSF and BPI (Fig. 7c). The light-evoked conductance of ChETA was similar in BPI and ACSF (Fig. 7c). Interestingly, when human neurons expressing ChETA-eYFP were patch-clamped in NEUMO and stimulated using the same light parameters (5, 100 ms intervals, at 475 nm), a significant reduction in action potential success rates was witnessed (Fig. 7d and Supplementary Fig. 10). Furthermore, the action potential light-evoked success rate, peak amplitudes, and the conductance of ChETA remained significantly lower in NEUMO compared to BPI despite increasing LED power (at 475 nm) (Fig. 7e).

Our patch-clamping experiments were limited to acute changes of basal media. To test the longer-term effects of the media with supplements, we repeated these experiments with iPSC-derived neurons on a multielectrode array recorder (MEA). We first recorded the spontaneous electrical activity of neurons in alternative media for 1 week (three media changes). Overall, despite occasional variability in the level of spontaneous firing, we found no significant difference in the spontaneous activity of the neurons in standard BrainPhys or BPI (Fig. 7f). In contrast, when neurons were switched to NEUMO for a week, the spontaneous activity dropped and did not recover over time unless it was switched back to BrainPhys medium (Fig. 7f, g). The human neurons were also transfected with an optogene (LV hSyn: ChETA-EYFP) and stimulated with blue light (475 nm) while on

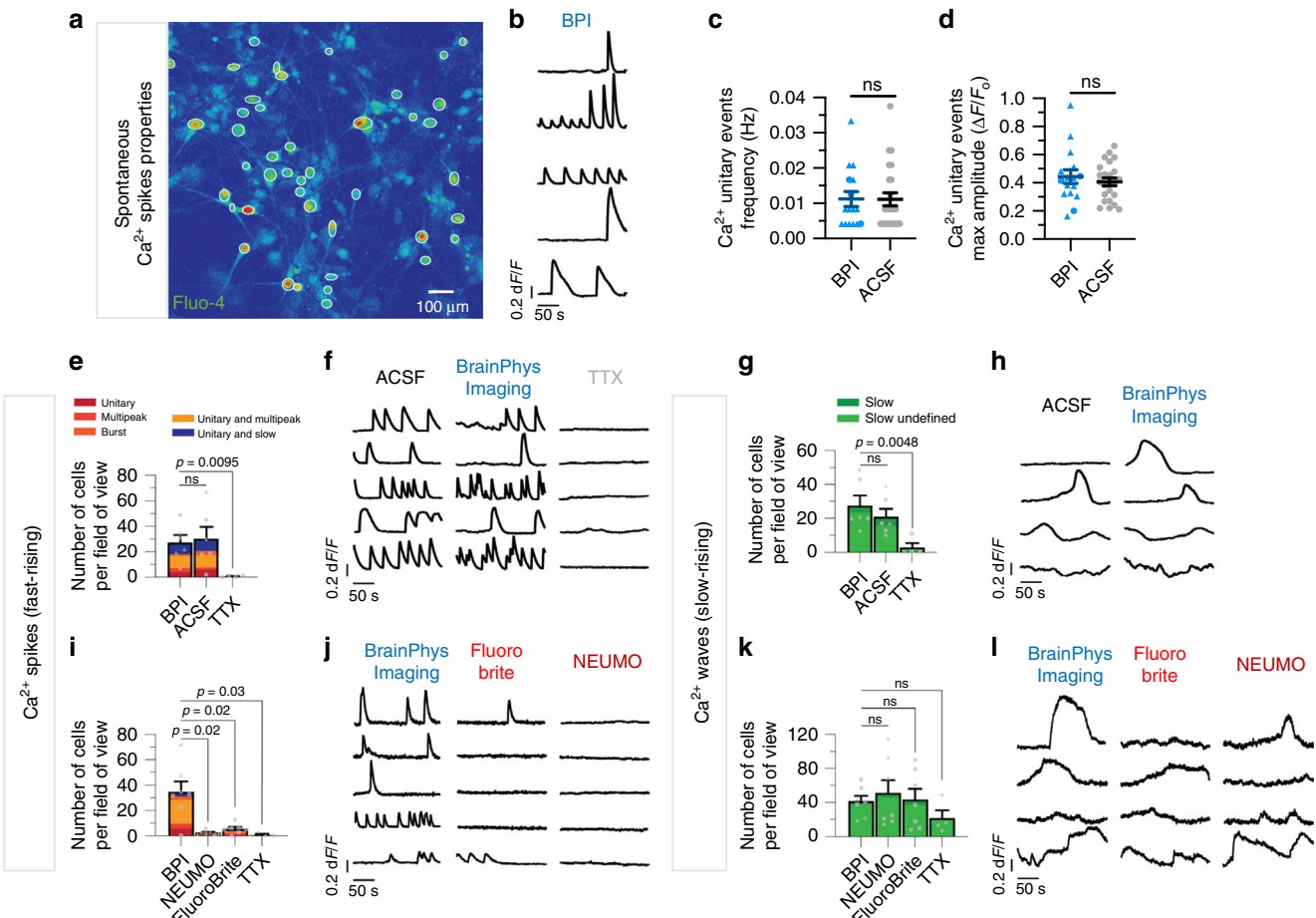

**Fig. 6 BrainPhys Imaging optimally supports fast-rising calcium spikes in human neurons in vitro.** Intracellular $Ca^{2+}$ changes in human PSC-derived neurons were measured with time-lapse image sequences of a $Ca^{2+}$ sensor (Fluo-4 AM). A total 1225 human neurons were analyzed across thirteen fields of view (FOVs) from four coverslips ($n = 4$ biologically independent experiments). Regions of interest (ROIs) were drawn on the cell soma to determine fluorescence intensity changes ($\Delta F/F_0$) over time. Active events ($\Delta F/F_0 > 5\%$ from baseline) were manually categorized into fast-rising $Ca^{2+}$ spikes or slow-rising $Ca^{2+}$ waves (see Fig. S8A). The same FOVs were imaged in artificial cerebrospinal fluid (ACSF), BPI, FluoroBrite, and NEUMO. **a** Example Fluo-4 AM fluorescence image of a neuronal population in BPI. White circles represent active ROIs. Fluorescent intensity represents intracellular $Ca^{2+}$ levels. **b** Typical fast-rising $Ca^{2+}$ spikes from five active cells in BPI. **c, d** No significant difference was observed between the average properties of fast-rising unitary $Ca^{2+}$ spike events in BPI ($n = 16$ cells) and ACSF ($n = 24$ cells) from six FOVs across two coverslips. **e, g** No significant difference was observed between the proportions of cells with fast $Ca^{2+}$ spikes or slow $Ca^{2+}$ waves in BPI and ACSF. The total active cells (BPI: $n = 329$; ACSF $n = 308$) were compared across six FOVs from two coverslips. Voltage-gated sodium channel blocker Tetrodotoxin (TTX, $1\,\mu M$) significantly reduced $Ca^{2+}$ spikes and waves. See also Supplementary Fig. 8a. **i, k** Comparison of $Ca^{2+}$ signals in BPI, NEUMO, and FluoroBrite show significant differences in the proportions of cells with $Ca^{2+}$ spikes but not waves. Active cells with $Ca^{2+}$ spikes and spikes/waves in BPI ($n = 243$ cells), FluoroBrite ($n = 39$ cells) and NEUMO ($n = 18$ cells) or with $Ca^{2+}$ waves in BPI ($n = 288$ cells), FluoroBrite ($n = 302$ cells), and NEUMO ($n = 356$ cells) were compared across seven FOVs from two coverslips. **f, h, j, l** Example traces from the same ROIs in different media. Symbols in **c–e, g, i, k** represent cells recorded first (triangles), second (circles), third (square), fourth (diamond) in either medium. Values are presented as mean ± SEM. Significance in **c–e, g, i, k** determined via two-tailed nonparametric unpaired (Mann–Whitney) tests. ns, $P > 0.05$.

the MEA recorder. Brief flashes of light were able to evoke action potentials in BPI but not in NEUMO (Fig. 7h, i). Altogether, this data demonstrates that BPI is optimal for optogenetic control of human neurons in vitro.

**BPI supports the viability and function of neuronal cultures.** Several components in the standard BrainPhys formulation were reduced or replaced to create BPI. Therefore, we wanted to determine whether the survival of neurons was compromised in BPI when culturing cells for extended periods of time. Both human and rat primary neurons cultured in BPI for 21 days exhibited healthy cell bodies and the expected neurite networks without apparent reduction in synaptic marker Synapsin1

(Fig. 8a). Importantly, the total number of live human or rat neurons in culture after 21 days in BPI was no different than in standard BrainPhys medium (Fig. 8b).

To determine if human neurons remain functional for extended periods in BPI, iPSC-derived neurons were tested on an MEA. After maturation for 6 weeks in standard BrainPhys medium, half the MEA plate was switched to BPI for a period of two weeks (Fig. 8c). Spontaneous mean firing rates and the proportion of active electrodes were similar between the wells kept in standard BrainPhys and those switched to BPI. Switching the cells back into the original BrainPhys after that period also did not affect the activity (Fig. 8c). Altogether, our data show that neuronal cultures are viable and functional when replacing standard BrainPhys for BPI with appropriate supplements for an

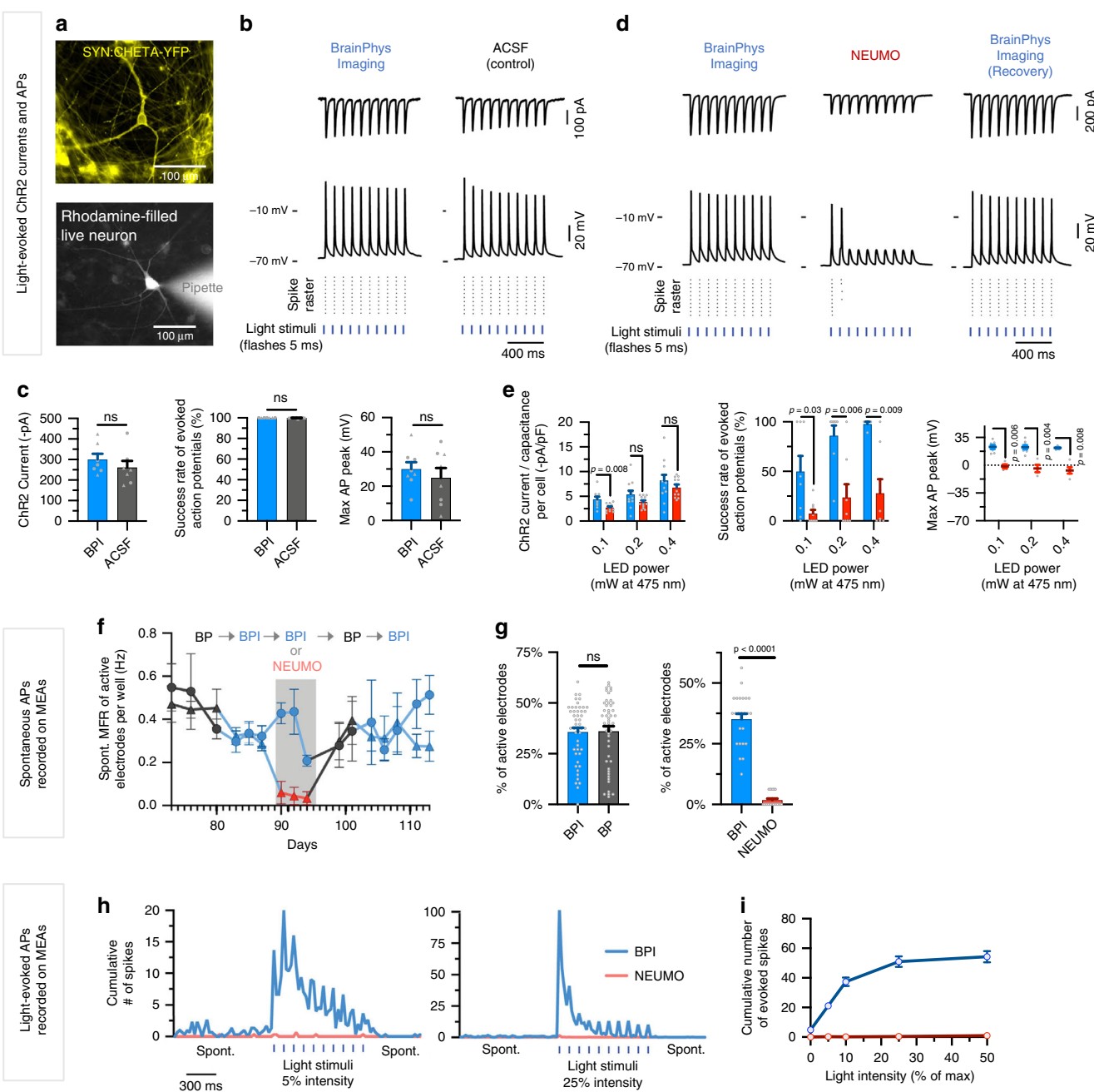

**Fig. 7 BPI supports the optogenetic control of human neurons in vitro. a–e** Optogenetics of human neurons with whole-cell patch-clamp recordings in BrainPhys Imaging (BPI), artificial cerebrospinal fluid (ACSF), and NEUMO. Each data point in the histograms represents a neuron. **a** Example images of neurons expressing synapsin:ChETA-YFP optogene and filled with rhodamine from the patch-pipette. **b** Whole-cell patch-clamp traces of neurons stimulated with 10 × 5 ms flashes of 0.4 mW blue light at 10 Hz in BPI or ACSF. Bottom of the graph shows corresponding spike raster plots. **c** Neurons patched in ACSF or BPI and stimulated with identical light parameters in **b** show no significant difference across ChR2 currents (BPI, ACSF; $n = 7$), action potential (AP) success rates (BPI, $n = 7$; ACSF, $n = 8$) and max AP amplitudes (BPI, ACSF; $n = 8$). Symbols represent neurons tested first (triangles) or second (circles) in either medium. **d** Optogenetic responses from the same patch-clamped neuron under 0.1 mW of blue light in BPI, NEUMO, and BPI (recovery) show optimal optogenetic control in BPI. **e** Quantification of patched neurons ($n = 22$ across four coverslips) recorded in BPI or NEUMO, stimulated with 10 × 5 ms flashes of blue light at 10 Hz with increasing power. **f–i** Optogenetically evoked and spontaneous firing rates of human neurons expressing synapsin:ChETA-YFP recorded in BP, BPI or NEUMO (with identical supplements) using multielectrode arrays (MEAs). Neurons were cultured in standard BP for 82 days before transitioning to "test" media. **f, g** Recordings were split into two groups: "circles'" (8 wells) and "triangles" (7 wells). Both groups were changed to BPI at day 82; only "triangles" switched to NEUMO at days 89–95. From day 96, both groups were cultured in BP for one week before switching to BPI. **h, i** Optogenetic responses of human neurons on MEAs in NEUMO ($n = 112$ electrodes) or BPI ($n = 128$ electrodes) when stimulated with blue light (Lumos) at increasing intensities. The cumulative number of spikes was summed in binning windows; **h** 27.5 ms bins; **i** 2 s bins from first light stimulus over three sweeps. Data in **h** were plotted with quadratic non-linear curves. Values in **c, e, g, i** are presented as mean ± SEM. Significance determined via two-tailed nonparametric unpaired (Mann–Whitney) tests. ns, $P > 0.05$.

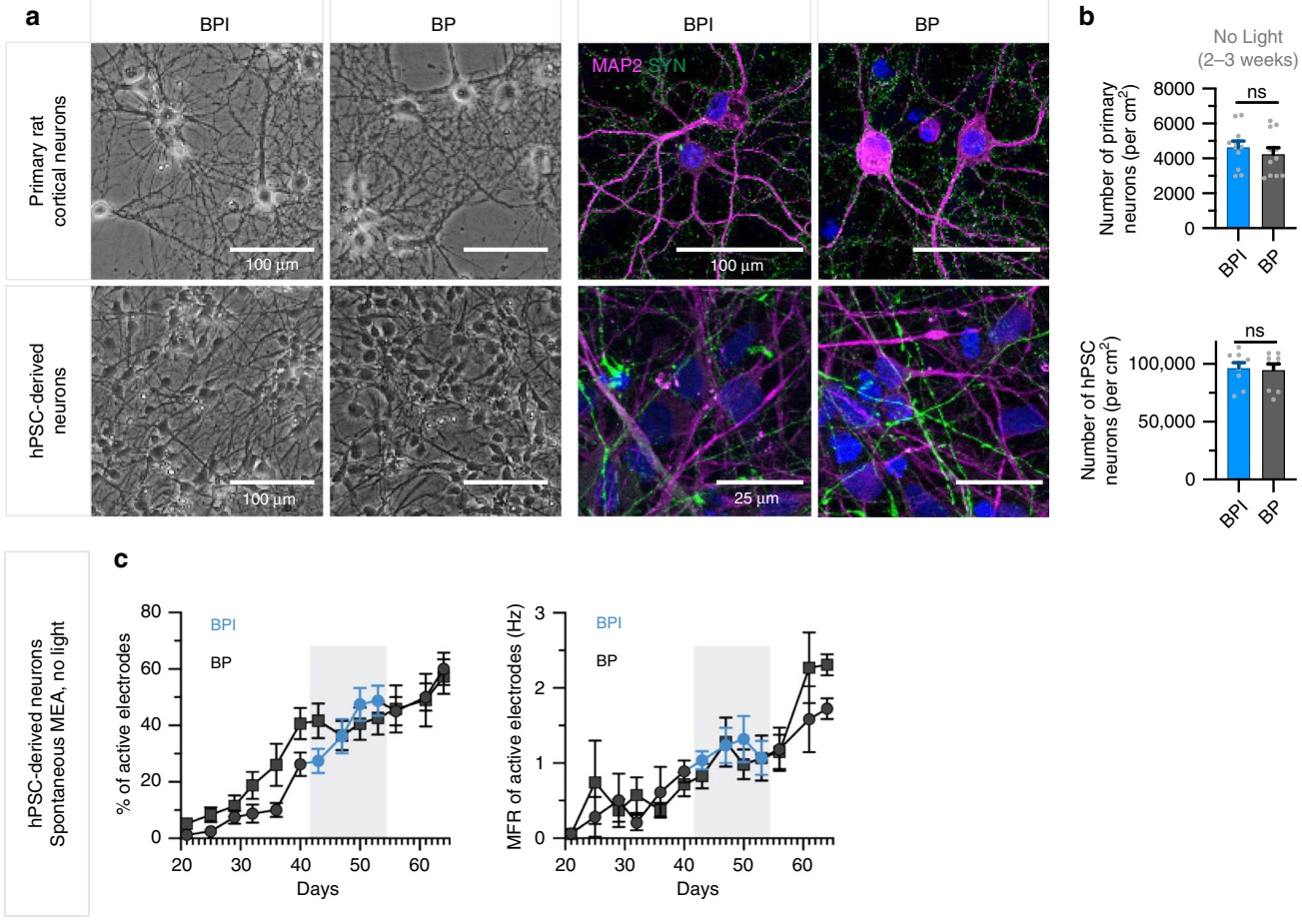

**Fig. 8 BrainPhys Imaging can support long-term viability and function in vitro. a** (left) Shows representative images of 14 days-in-vitro (DIV) rat cortical primary neurons and hPSC-derived neurons matured in BrainPhys (BP) and BrainPhys Imaging (BPI) exhibiting healthy neuron morphology. **a** (right) Representative immunocytochemistry images showing that 21 DIV primary rat cortical neurons and 14 DIV hPSC-derived neurons characterized by MAP2 expression (magenta) matured in BP or BPI also display an appropriate expression of synaptic marker Synapsin1 (green). **b** Quantification of the number of neurons per cm² as represented in **a** shows that BPI supports equivalent neuronal survival relative to BP for 21 DIV rat cortical primary neurons ($n = 11$) and 14–21 DIV hPSC-derived neurons ($n = 9$). Values represent mean ± SEM. Significance determined using two-tailed nonparametric unpaired (Mann–Whitney) tests; ns $P > 0.05$. **c** Multielectrode array (MEA) recordings from hPSC-derived neurons cultured in a 48-well MEA plate in BP for 9 weeks ($n = 6$ wells, 96 electrodes) or switched to BPI from BP during weeks 6–8 ($n = 5$ wells, 80 electrodes) represent the spontaneous mean firing rates and percentage of active electrodes (>0.017 Hz) for each data set. The percentage of active electrodes and mean firing rates were both maintained during the period in BPI. Values are presented as mean ± SEM.

extended period (2–3 weeks). For longer time periods, it is possible to alternate the feed with standard BrainPhys to replenish the light-sensitive nutrients reduced in BPI.

## Discussion

BPI basal medium was designed to enhance imaging capacity and support neuronal function in vitro. BPI proved advantageous for light-driven analysis of cultures of reprogrammed human brain cells and primary neurons. The most commonly used fluorophores in cell biology have a spectrum of excitation and emission from ultraviolet (~300 nm) to infrared. From ultraviolet to infrared, our data show that BPI optimizes the absorbance, autofluorescence, signal-to-background ratio, and phototoxicity, and outperforms other tissue culture media. Our data also shows that BPI supports physiological action potentials, synaptic activity, calcium spikes, and optogenetics function while supporting the long-term viability required for neuronal cultures.

We have shown that BPI can be used with multiple neuronal cell types in vitro, including human and rat, primary or reprogrammed, cortical or midbrain, 2D or 3D. Although we could not

test all the possible experimental designs, we predict that most brain cell models in vitro or ex vivo should benefit from BPI in light-driven experiments. No experiment that we performed indicated that BPI might be limited to a particular neuronal cell type. Any cell type supported by standard BrainPhys will benefit from switching temporarily to BPI during light-driven experiments. Patient-derived iPSCs are increasingly popular for preclinical models of neurology[28–39]. BrainPhys medium and the imaging formula presented here were specifically designed with the intent to improve such human neuronal models.

We do not foresee any particular limitation in the type of live imaging experiment benefiting from BPI compared to the other neuronal media tested in vitro. The improvement in fluorescent signals and minimal photocytotoxicity in BPI can help most light-driven neuronal experiments, in particular, the ones with low signals and requiring higher light stimulation power[18]. In addition, the optimal support of electrophysiological activity is necessary for light-driven technologies used to monitor neuronal function (e.g., voltage-sensors, calcium-sensors, synaptic imaging, pH-sensors, and mitochondria motility)[2,4–8,10]. The discovery of personalized treatment with patient-derived neurons will benefit

from drug screening in vitro[40,41]. Most drug screens use high-throughput microscopy to assess cell survival, which can be confounded by phototoxic media. Our data suggest that BPI will be the medium of choice for live-cell high-content imaging experiments in neurological and psychiatric research.

In the last decade, optogenetics has become a major research tool to interrogate the function of nerve cells in health and disease[2,4,7,42,43]. The integration of optogenetics with neural stem cells has enabled selective excitation and/or inhibition of specific cell populations expressing optogenes with exceptional precision, allowing researchers to investigate neural network function from innovative perspectives[44–48]. As BPI reduces photo-toxicity and optimizes electrical and synaptic activity, such conditions certainly improve optogenetics ability to assess neural networks. We have demonstrated that BPI is highly compatible with modulating the activity of human neurons in vitro with light excitation of ultrafast channel rhodopsin (ChETA). Similarly, BPI may also become advantageous for other optical experiments such as the photorelease of caged compounds[13,14], or optical activation of intracellular antibodies[15].

Standard BP with appropriate supplements and tissue culture techniques is capable of sustaining neuronal maturation for over 6 months in vitro[26]. In BPI, several components were reduced, removed or replaced from the standard BP to improve imaging capacity. We have shown that despite these modifications, BPI is capable of supporting cell viability and function (human neurons or rat primary neurons) for at least several weeks to similar levels observed in standard BrainPhys. However, BPI was designed to complement standard BrainPhys rather than replace it completely. It is hard to predict the viability effect of a medium on every cell type used in neuroscience research. It is possible that some cell types might be more sensitive than others. In some cases, it may be best to culture the cells in standard BrainPhys for the majority of the time and only switch to BPI when required for imaging assays. For longer-term imaging experiments of particularly sensitive cells, it is also possible to replenish the cells with standard BP every week or two. Despite this note of caution, based on our experiments, we are confident that using BPI for optical assays (short-term and long-term) will be better than using standard neuronal media. It is also important to note that like any other basal media, long-term viability requires the addition of appropriate supplements. In this study, we only tested the supplements generally recommended with standard BrainPhys (SM1 with or without Retinoic acid, N2-A, BDNF, GDNF, ascorbic acid, laminin, cAMP, and IGF-I), but it may be possible to obtain similar results with different sets of supplements if they are not phototoxic.

To surpass reported light-induced damage on cells cultured in Neurobasal or DMEM, two photostable basal media were developed: BrightCell™ NEUMO and FluoroBrite™ DMEM[25]. Our analysis confirmed some improvements in the autofluorescence of these media compared to standard media (Neurobasal, DMEM/F12, or standard BP). However, both media still exhibited mild autofluorescence and increased absorbance in shorter wavelength light, whereas the autofluorescence of BPI was significantly reduced throughout a wider light spectrum (300–800 nm). We do not know the cause of the residual fluorescence at shorter wavelengths in the other tissue culture media specialized for imaging, but it is likely that not all photoactive components were adjusted appropriately in these formulations. The optical superiority of BPI compared to other imaging media may be particularly prominent for experiments requiring peak excitation between 300 and 400 nm, such as blue fluorophores (e.g., DAPI, Hoechst, eBFP, Brilliant Ultraviolet, DyLight 405), calcium sensors (e.g., Fura-2), or other cellular sensors (e.g., Lysotracker Blue/MeOH). Compared to standard media such as BrainPhys,

we observed significant increases in signal-to-background ratios in BPI with blue and green fluorophores. The prominence of the improvements may vary depending on the quality of the fluorophore and imaging system (e.g., lens magnification, numerical aperture values). No major differences were found in imaging red fluorophores in BPI and BP, which is likely attributed to the low absorbance and autofluorescence for long wavelengths (Fig. 1b, c). NEUMO was specifically designed for the culture of neuronal cells and to replace the Neurobasal medium during live-cell imaging[25]. However, when tested on human neurons in vitro, it failed to support spontaneous calcium spikes or optogenetic-induced action potentials. In general, the electrophysiological activity in NEUMO was substantially reduced compared to side-by-side recordings in ACSF or BrainPhys media. This result was consistent with multiple electrophysiological techniques including patch-clamping, calcium imaging, and MEA on human PSC-derived neurons. It also aligns with previous data obtained comparing the electrophysiological activity of human PSC neurons in standard BrainPhys and Neurobasal[26]. FluoroBrite DMEM is another photostable medium developed to replace DMEM in experiments requiring light stimulation. Like standard DMEM, FluoroBrite was not designed for neuronal cultures. Live calcium imaging data suggest that FluoroBrite is suboptimal for neuronal culture function. However, it is possible that it is better suited for other cell types. The suboptimal calcium spikes observed in NEUMO and FluoroBrite media may be related to their formulations driving higher calcium store release, and consequently suppressing neuronal activity by activating calcium-gated potassium currents and increasing action potential thresholds[49]. In a blind comparative analysis, our data show that BPI surpasses alternative photostable basal media when used on functional human PSC-derived neuronal culture.

Optically driven technology is known to induce phototoxic by-products in culture, leading to cellular death and metabolic alterations after light exposure. Furthermore, currently available imaging media, which were designed to limit phototoxicity, impair electrophysiological activity, and fail to support fundamental neurophysiological properties. The design of BPI is timely with the rapid expansion of functional imaging and human neuronal models in vitro. BPI, which is a reformulation of the standard BrainPhys basal medium, is found to improve optical properties throughout the entire light spectrum required for fluorophores and sensors used in cellular biology. The optically optimized properties of BPI will enhance the quality of results for a plethora of light-driven experiments in vitro. Such technical advancement provides a useful tool for preclinical models in vitro and may, in turn, facilitate translation in neurology and psychiatry.

## Methods
All catalog numbers for reagents, antibodies, biological samples, and additional resources can be found in Supplementary Table 1 with corresponding source details.

**Human pluripotent stem cell (hPSC)-derived neuron culture**. Human neuron cultures were derived from pluripotent stem cells in two independent laboratories using slightly different protocols. Method #1 (Figs. 2–7 and Supplementary Figs. 2, 3, 6–10): WA09 (H9) ES cell colonies (Lot No. WB66595, WiCell) were maintained in mTeSR™1 (Cat. No. 85850, STEMCELL Technologies) as per manufacturer's instructions on cell culture ware coated with hESC-qualified Matrigel (Cat. No. 356277, Corning). The cells were split by dispase treatment (Cat. No. 07923, STEMCELL Technologies) and mechanical scraping every 3–5 days onto fresh Matrigel. Embryoid body-based neural induction was performed as on a previous protocol[26,50] with modifications. Briefly, hESCs were cultured in an ultralow attachment plate in a neural induction medium (NIM) consisting of DMEM/F12 with 15 mM HEPES (Cat. No. 36254, STEMCELL Technologies), 1× NeuroCult SM1 (Cat. No. 05711, STEMCELL Technologies), 1× N2 Supplement-A (N2-A; Cat. No. 07152, STEMCELL Technologies), 10 μM SB431542 (Cat. No. 72232, STEMCELL Technologies) and either 500 ng/ml Noggin (120-10C, PeproTech) or

100 nM LDN193189 (Cat. No. 72147, STEMCELL Technologies). Media was exchanged every other day for 7 days to allow embryoid bodies (EBs) formation. EBs were transferred to a tissue culture dish coated with 10 μg/ml poly-L-ornithine (Cat. No. P3655, Sigma) and 5 μg/ml laminin (Cat. No. 23017015, Thermo Fisher Scientific) in NIM supplemented with 1 μg/ml laminin. Media was exchanged every other day for 7 days until neural rosettes were clearly visible. Neural rosettes were manually selected under phase contrast (EVOS XL Core Imaging System, Thermo Fisher Scientific) and transferred onto fresh Matrigel-coated plates in neural progenitor medium (NPM). Midbrain neural progenitor differentiation: Midbrain neural progenitor medium (mNPM) was composed of DMEM/F12+ GlutaMAX (Cat. No. 10565018, Thermo Fisher Scientific) supplemented with 1× SM1, 1× N2-A, 200 ng/ml Sonic Hedgehog (Cat. No. 100-45, PeproTech), 100 ng/ml FGF8b (Cat. No. 100-25, PeproTech), 200 nM ascorbic acid (Cat. No. A4403, Sigma) and 1 μg/ml laminin. Cortical neural progenitor differentiation: Cortical neural progenitor medium (cNPM) consisted of DMEM/F12+ GlutaMAX plus 1× SM1, 1× N2-A, 10 ng/ml FGF2 (Cat. No. 78134, STEMCELL Technologies), 200 nM ascorbic acid and 1 μg/ml laminin. Neuronal progenitor cells (NPCs) were maintained at high density, fed every other day with fresh NPM and split about once a week at 1:3–1:4 ratio onto Matrigel-coated plates using Accutase (Cat. No. 07920, STEMCELL Technologies). For neuronal maturation, NPCs were dissociated and seeded at a density of $8 \times 10^4$ to $2 \times 10^5$ cells/cm² (cell line dependent) onto tissue culture plates coated with 10 μg/ml poly-L-ornithine and 5 μg/ml laminin in neural maturation medium (NMM). Midbrain neuronal maturation medium ("mNMM") was composed of BrainPhys™ (Cat. No. 05790, STEMCELL Technologies) supplemented with 1× SM1, 1× N2-A, 20 ng/ml BDNF (Cat. No. 78133.1, STEMCELL Technologies), 20 ng/ml GDNF (Cat. No. 78139.1, STEMCELL Technologies), 0.5 mM dibutyryl cyclic-AMP (Cat. No. D0627, Sigma), 200 nM ascorbic acid and 1 μg/ml laminin. Cortical neuronal maturation medium ("cNMM") was composed of BrainPhys™ supplemented with 1× NeuroCult™ SM1 without vitamin A (Cat. No. 05731, STEMCELL Technologies), 1× N2-A, 20 ng/ml BDNF, 20 ng/ml GDNF, 10 ng/ml IGF-I (Cat. No. 78022.1, STEMCELL Technologies), 0.5 mM dibutyryl cyclic-AMP, 200 nM ascorbic acid and 1 μg/ml laminin. Half media changes with NMM were performed every 2–3 days. To improve neuronal culture quality and viability and achieve a more even distribution of cells for final neuronal maturation, postmitotic neuronal cultures 14 days postmaturation were routinely sorted (BD FACSMelody cell sorter, BD Biosciences) and live (DAPI negative) cells were replated at a density of $1.6–2.1 \times 10^5$ cells/cm² (cell line dependent) on poly-L-ornithine-coated and laminin-coated glass coverslips or tissue culture-treated plates in NMM containing half concentration of growth factors.

Method #2 (Fig. 8a, b): frozen neural progenitor cells (NPCs) derived from embryonic (ES) or induced pluripotent (iPS) cells (XCL-1-derived neural stem cells, STiPS-M001 induced pluripotent stem cells, H14-derived neural stem cells and STiPS-B004 induced pluripotent stem cells[51]) were differentiated into neuronal precursors with the STEMdiff™ Neuron Differentiation Kit (Cat. No. 08500, STEMCELL Technologies). Neuronal precursor cells were then frozen with Cryostor CS10 (Cat. No. 07930; STEMCELL Technologies) in liquid nitrogen storage and thawed later for experiments. Post-thaw, cells were plated in 500 μl per well of STEMdiff™ Neuron Differentiation medium at 26,000–130,000 cells per cm² (cell line dependent) in a 24-well plate. On day 1 post-thaw, 500 μl of BrainPhys™ (Cat. No. 05790, STEMCELL Technologies) or BrainPhys™ Imaging (Cat. No. 05796, STEMCELL Technologies) supplemented with 1× NeuroCult™ SM1, 1× N2-A, 20 ng/ml BDNF (Cat. No. 78005, STEMCELL Technologies), 20 ng/ml GDNF (Cat. No. 78058, STEMCELL Technologies), 200 nM ascorbic acid (Cat. No. 72132, STEMCELL Technologies), 1 μg/ml laminin (Cat. No. L2020, Sigma) and 0.5 mM dibutyryl cyclic-AMP (Cat. No. 73882, STEMCELL Technologies) [complete BrainPhys™] was added per well. Half media changes were performed every 3–4 days with complete BrainPhys™ or BrainPhys™ Imaging.

**Neuronal Basal Media Tested**. ACSF contained (in mM) 121 NaCl (Cat. No. S6191, Sigma), 4.2 KCl (Cat. No. 60128, Sigma), 1.1 CaCl₂ (Cat. No. 21097, Sigma), 1 MgSO₄ (Cat. No. 63138, Sigma), 29 NaHCO₃ (Cat. No. S5761, Sigma), 0.45 NaH₂PO₄–H₂O (Cat. No. S5011, Sigma), 0.5 Na₂HPO₄ (Cat. No. S5136, Sigma) and 20 glucose (Cat. No. G7021, Sigma). BrainPhys™ (Cat. No. 05790, STEMCELL Technologies), BrainPhys™ Imaging (Cat. No. 05796, STEMCELL Technologies), BrainPhys without Phenol Red (Cat. No. 05791, STEMCELL Technologies), FluoroBrite™ DMEM (Cat. No. A1896701, Thermo Fisher Scientific), DMEM/F12 (Cat. No. 10565018, Thermo Fisher Scientific), Neurobasal (Cat. No. 21103-049, Thermo Fisher Scientific), and BrightCell™ NEUMO (Cat. No. SCM146, Merck) were obtained via the source and catalog details provided. For long-term maturation of human neuronal culture, all basal media were supplemented with:

Human midbrain neurons supplements: 1× N2-Supplement A (Cat. No. 07152, STEMCELL Technologies), 1× NeuroCult™ SM1 (Cat. No. 05711, STEMCELL Technologies), 200 nM ascorbic acid (Cat. No. A4403, Sigma), 20 ng/ml or 10 ng/ml BDNF (Cat. No. 78133.1, STEMCELL Technologies), 20 ng/ml or 10 ng/ml GDNF (Cat. No. 78139.1, STEMCELL Technologies), 1 μg/ml laminin (Cat. No. 23017015, Thermo Fisher Scientific), and 0.5 or 0.25 mM dibutyryl cyclic-AMP (Cat. No. D0627, Sigma). Human cortical neurons supplements: 1× N2-Supplement A (Cat. No. 07152, STEMCELL Technologies), 1× NeuroCult™ SM1 without vitamin A (Cat. No. 05731, STEMCELL Technologies), 200 nM ascorbic acid (Cat. No. A4403, Sigma), 20 or 10 ng/ml BDNF (Cat. No. 78133.1, STEMCELL

Technologies), 20 or 10 ng/ml GDNF (Cat. No. 78139.1, STEMCELL Technologies), 10 or 5 ng/ml IGF-I (Cat. No. 78022.1, STEMCELL Technologies), 1 μg/ml laminin (Cat. No. 23017015, Thermo Fisher Scientific), and 0.5 or 0.25 mM dibutyryl cyclic-AMP (Cat. No. D0627, Sigma). Two concentrations are listed for growth factors and dibutyryl cyclic AMP since these factors were reduced by 50% in the media after 2 weeks of neuronal maturation.

**Generation of GFP-expressing human pluripotent stem cell line**. WA01 (H1) embryonic stem cells (WiCell) were cultured in mTeSR™1 as per the manufacturer's instructions (Cat. No. 85850, STEMCELL Technologies) plated on Matrigel™ (Cat. No. 356277, Corning). A stable GFP overexpressing cell line was generated by transfecting cells with a plasmid encoding GFP expression driven by the chicken beta-actin promoter[52,53] using the Neon Transfection System (Thermo Fisher Scientific). H1 cells were dissociated to single cells and $1.2 \times 10^6$ cells were resuspended in 120 μl of R-buffer containing 2 μg of pCAG:GFP:IRES:Puro. Cells were then electroporated using the manufacturer's instructions with the following settings: 1200 volts, 20 ms, 1 pulse. The transfected cell suspension was then added to 4 ml of mTeSR™1 supplemented with CloneR™ (Cat. No. 05888, STEMCELL Technologies), mixed by pipetting, then added to tissue culture-treated plates coated with Matrigel™ at $2.5 \times 10^4$ cells/cm². The medium was replaced after 24 h and fed with fresh medium daily until the cells were ~60% confluent. The cells were then fed with fresh medium containing 1 μg/ml puromycin (Cat. No. 13844, Cayman Chemicals) daily to select for stable transfectants. Clonal cell lines were then generated by dissociating the H1-GFP cells to single cells and seeding at 10 cells/cm² in mTeSR™1 supplemented with CloneR™ onto tissue culture-treated plates coated with Matrigel™. Cells were then fed 2 days post-plating with fresh medium supplemented with CloneR™, then fed daily following 4 days post-plating with mTeSR™1. Colonies were then picked manually between 7–9 days and cultured as outlined in "Human Pluripotent Stem Cell (hPSC)-derived Neuron Culture – Method #2".

**Generation of forebrain organoids**. The dorsal forebrain organoids and ventral forebrain organoids were generated using the STEMdiff™ Dorsal Organoid Kit (Cat. No. 08620, STEMCELL Technologies) and STEMdiff™ Ventral Organoid Kit (Cat. No. 08630, STEMCELL Technologies) respectively[54,55]. Dorsal forebrain organoids were generated from unlabeled WA01 (H1) embryonic stem cells while the ventral forebrain organoids were generated from GFP-labeled WA01 (H1) embryonic stem cells. On day 0, hPSCs were washed with PBS and detached using Gentle Cell Dissociation Reagent (Cat. No. 07174, STEMCELL Technologies) for 8 min at 37 °C. Cells were resuspended in DMEM/F12 and centrifuged at 300×g for 5 min at room temperature. The supernatant was removed and hPSCs were then resuspended in Forebrain Organoid Formation Medium containing 10 μM Y-27632 (Cat. No. 10005583, Cayman Chemical) to a concentration of $3 \times 10^6$ cells/ml. An AggreWell™800 plate (Cat. No. 34811, STEMCELL Technologies) was prepared by pre-treating a well with 500 μl of Anti-Adherence Rinsing Solution (Cat. No. 07010, STEMCELL Technologies) followed by brief centrifugation at 2000×g for 5 min at room temperature, Anti-Adherence Rinsing Solution was removed by aspiration and replaced with 1 ml of Forebrain Organoid Formation Medium containing 10 μM Y-27632. To each AggreWell™800 well, 1 ml ($3 \times 10^6$ cells) of cell suspension was added, and the plate was centrifuged at 100×g to capture cells into microwells. The plate was grown in an incubator at 37 °C and 5% CO₂. From day 1–5, media was changed daily using partial medium changes (1.5 ml/well) using Forebrain Organoid Formation Medium. On day 6, neural aggregates were harvested using a wide-bore pipette tip to transfer aggregates onto a 37 μm reversible strainer (Cat. No. 27250, STEMCELL Technologies) to remove single cell debris. Organoids were added to a 6-well suspension culture plate (Cat. No. 27145, STEMCELL Technologies) with Forebrain Organoid Expansion Medium, approximately 25–50 forebrain organoids were distributed per well of the 6-well plate. For ventral forebrain organoids, STEMdiff™ Neural Organoid Supplement D (Cat. No. 08631, STEMCELL Technologies) was added to the Forebrain Organoid Expansion Medium. From day 6–24, full media exchange was performed every two days using Forebrain Organoid Expansion Medium for the dorsal forebrain organoids or Forebrain Organoid Expansion Medium containing STEMdiff™ Neural Organoid Supplement D for the ventral forebrain organoids. On day 25, the media was replaced with Forebrain Organoid Differentiation Medium for both the dorsal forebrain organoids and ventral forebrain organoids, with full media exchange performed every two days. At day 30, a single dorsal forebrain organoid and a single ventral forebrain organoid were removed from suspension culture using a wide-bore pipette tip and were placed together into one well of a 96-Well U-bottom plate (Cat. No. 7007, Corning) in 200 μl of Forebrain Organoid Differentiation Medium. The forebrain organoids were fed every two days using half-media exchange (100 μl per well) and were allowed to form an assembloid over one week.

**Primary rat neuron culture (Figs. 2e–f, 3a–d, k, 8a–b and Supplementary Figs. 1c–f, 4b, c, 5)**. Pairs of Rat E18 cortices (Cat. No. SDECX, BrainBits, LLC) were dissociated for 10 min in papain (Cat. No. LK003176; Worthington Biochemical; at least 20 U/ml). A single-cell suspension was obtained and filtered through a 40 μm cell strainer. The resulting primary cells were cultured in

NeuroCult™ Neuronal Plating medium (Cat. No. 05713, STEMCELL Technologies) or Neurobasal medium for Neurobasal and NEUMO cultures (Cat. No. 21103-049, Thermo Fisher Scientific) with 1× SM1 (Cat. No. 05711, STEMCELL Technologies), 0.5 mM L-glutamine (Cat. No. 07100, STEMCELL Technologies) and 25 μM L-glutamic acid (Cat. No. G8415, Sigma) on culture-ware pre-coated with 10 μg/ml poly-D-lysine (Cat. No. P7280, Sigma). Cells were plated at 30,000 cells/cm$^2$ in a 24-well plate. Five days post-plating, half media changes were performed every 3–4 days with either BrainPhys™ (Cat. No. 05790, STEMCELL Technologies), Neurobasal, BrightCell™ NEUMO (Cat. No. SCM146, Merck) BrainPhys™ without Phenol Red (Cat. No. 05791, STEMCELL Technologies), or BrainPhys™ Imaging (Cat. No. 05796, STEMCELL Technologies) supplemented with 1× SM1. Neurobasal and BrightCell™ NEUMO media were also supplemented with 0.5 mM L-glutamine.

**Osmolality measurements.** To assess media osmolality, 20 μl of test media or supernatant collected from maturing human iPSC-derived cortical neurons before feeding was pipetted into a disposable tube (Cat. No. 22-046733, Advanced Instruments). A Fiske Micro-Osmometer, model 210 (Cat. No. 14-727-420, Advanced Instruments) was used and calibrated with 2000, 850, 290, and 50 mOsmol/kg standards. Before recordings, the calibration was referenced against 290 mOsmol/kg standards, and between recordings, the micro-osmometer probe was cleaned with a probe cleaner (Cat. No. 3MA800, Advanced Instruments). Three human cerebrospinal fluid (hCSF) samples were pooled from multiple subjects (1–4).

**Absorption spectra of culture media.** Absorption spectra were measured using a CARY 7000 spectrophotometer (Agilent Technologies) with an integrating sphere attachment and a cuvette center mount holder designed to measure the absorbance of liquid samples[56]. A 1 cm pathlength cuvette with all four clear sides was used with 3 ml of sample for each measurement and spectra were measured from 200 to 800 nm in 5 nm increments with a spectral bandwidth of 2 nm. Acquired spectra were baseline corrected and blank subtracted to obtain corrected absorption spectra.

**Autofluorescence measurements of culture media.** Media autofluorescence was tested in two independent laboratories using slightly different protocols and apparatus. Method #1 (Fig. 1c): Volumes (10 ml) of control and test media were aliquoted into a blacked-out 15 ml tube (Corning). A custom bifurcated fiber probe (200 μm, 0.22 NA) was dipped into the media. 375, 405, 488, and 532 nm excitation wavelengths (20 mW) were sequentially fired from a Toptica iCHROME MLE (Farmington). Emission spectra (400–700 nm) from each excitation wavelength were captured using a Horiba MicroHR (Kyoto, Japan) using a 500 ms integration time. The following long-pass filters were used for 375, 405, 488, and 532 nm generated spectra respectively: 405 nm (Cat. No. LP02-405RU-25, Semrock Razoredge), 488 nm (Cat. No. LP02-488RE-25, Semrock Razoredge), 532 nm (Cat. No. LP03-532RE-25, Semrock Razoredge). Method #2 (Fig. 1d–f): Equal volumes (300 μl) of control and test media were pipetted into a CellCarrier-96 black plate with an optically clear bottom (Cat. No. 60055550, Perkin Elmer). A total of eight replicate wells per medium were tested with a BMG Labtech FLUOstar Omega using an excitation/emission of 355/460 (blue), 485/520 (green), 544/590 (red) and 584/620 nm (far-red). FLUOstar uses a high energy xenon flash lamp light source with a 100 Hz flash rate and 167 ms exposure time. The energy per flash emitted is 67 mJ and subjects each well to 1 mW of light power. Emission wavelengths were captured using a BMG Labtech photomultiplier tube detector (PMT) at a gain of 1000. Blue, green, red, and far-red filters utilized the respective bandwidths for their excitation/emission wavelengths: 20/20, 12/25, 20/20, and 10/20 nm. Acquired autofluorescence data was blank subtracted. For normalization, the mean fluorescence intensity in PBS was subtracted from the other media.

**Lentiviral vectors.** Lentiviral vectors were produced in Lenti-X™ 293T cells (Cat. No. 632180, Takara Bio) cultured in DMEM, high glucose, no glutamine (Cat. No. 11960044, Thermo Fisher Scientific) supplemented with 10% fetal bovine serum (Cat. No. 10-099-141, Thermo Fisher Scientific), 4 mM GlutaMAX supplement (Cat. No. 35050061, Thermo Fisher Scientific) and 1 mM sodium pyruvate (Cat. No. S8636, Sigma) on cell culture-ware coated with 0.002% poly-L-lysine (Cat. No. P7280, Sigma). Lenti-X™ 293T cells were transfected with 12.2 μg lentiviral transfer plasmid (LV Syn-hChR2-T159C-E123T-EYFP-WPRE [hSyn:ChETA-YFP[11]], or LV pCSC-Synapsin(0.5 kb)-MCS-EGFP, a derivative of pCSC-SP-PW-EGFP) and packaging plasmids (8.1 μg pMDL/RRE, 3.1 μg pRSV/REV, and 4.1 μg pCMV-VSVg) using a polyethylenimine (PEI, 25 kDa, linear; Cat. No. 23966-1, Polysciences) transfection method (4:1 ratio of PEI to DNA). The culture medium was exchanged six hours after transfection. The supernatant containing lentiviral particles was collected ~66 h after transfection, filtered through a sterile 0.45-μm SFCA membrane filter (Cat. No. 431200, Corning) and ultracentrifuged at 25,000 rpm at 4 °C for 2 h. Virus pellets were resuspended in Hank's Balanced Salt Solution (Cat. No. 14025092, Thermo Fisher Scientific) and virus titers were determined using the Lenti-X™ qRT-PCR Titration Kit (Cat. No. 632180, Takara Bio) according to the manufacturer's instructions. Neuron cultures were transduced with titer-matched lentiviral vectors (2.67 × 10$^3$ viral RNA copies/cell) for 48 h prior to wash.

Lentiviral transfections in neuronal cultures were performed at least one week prior to experimentations. If expression levels were low, neurons were left in culture for an extra 3–5 days before further experimentations commenced.

**Immunofluorescent staining protocol.** Cultures were fixed in 4% paraformaldehyde (Cat. No. J61899, Alfa Aesar) at 22 °C for 30 min. Cells were then permeabilized with 0.1% Tween-20 (Cat. No. P7949, Sigma) in PBS (Cat. No. 37350. STEMCELL Technologies)) for 15 min. Cells were incubated overnight at 4 °C with anti-β-III-tubulin antibody (Cat. No. 801201, Lot No. B264428, Biolegend; 1:1000), anti-MAP2 antibody (Cat. No. ab5392, Lot No. GR3265288-3, Abcam; 1:5000) and/or anti-Synapsin I antibody (Cat. No. ab1543, Lot No. 2930537, Merck; 1:2000) diluted in PBS containing 10% normal donkey serum (Cat. No. S30, Merck). The next day cells were washed three times with PTW and incubated with antimouse Alexa Fluor®488 (Cat. No. 715-545-150, Lot No. 132977, Jackson Immunoresearch; 0.625 μg/ml), antirabbit Alexa Fluor®488 (Cat. No. 711-545-152, Lot No. 116141, Jackson Immunoresearch; 0.625 μg/ml), anti-chicken Alexa Fluor®488 (Cat. No. 703-545-155, Lot No. 116967, Jackson Immunoresearch; 0.625 μg/ml), antichicken Alexa Fluor®647 (Cat. No. 703-605-155, Lot No. 114163, Jackson Immunoresearch; 0.625 μg/ml) and/or antimouse DyLight®594 (Cat. No. 35510, Lot No. M163715, Thermo Fisher Scientific; 1 μg/ml) diluted in PBS containing 2% normal donkey serum overnight at 4 °C. Cells were then washed three times with PTW to remove residual antibodies. To detect the cell bodies, Hoechst 33342 (Cat. No. 14533, Lot No. 056M4091V, Sigma; 2 μg/ml) was used to counterstain the cells.

**Neurofluor™ NeuO labeling protocol.** E18 rat cortical neurons were cultured in BrainPhys™ (Cat. No. 05790, STEMCELL Technologies), BrainPhys™ Imaging (Cat. No. 05796, STEMCELL Technologies) or BrainPhys without Phenol Red (Cat. No. 05791, STEMCELL Technologies) for 11 days (16 days total) before labeling with NeuroFluor™ NeuO (Cat. No. 01801, STEMCELL Technologies). For labeling, the culture medium was aspirated and replaced with a labeling medium consisting of culture medium plus 0.25 μM NeuroFluor™ NeuO. Cells were incubated in the labeling medium for 1 h at 37 °C. After incubation, the labeling medium was replaced with fresh medium and cells were incubated at 37 °C for 2 h before imaging.

**Image acquisition.** Either an SP8 confocal microscope (Leica), ImageXpress Micro 4 High Content Screening System (Molecular Devices) with an Andor SDK3 camera, CKX53 inverted microscope with an SC100 digital camera (Olympus), or a BX51 upright microscope (Olympus) with a PCO.Panda 4.2 digital camera were used for image acquisition. All images were taken using 16-bit cameras. Varying lens parameters and acquisition settings were used, as follows:

**Imaging fixed primary rat cortical neurons (Figs. 2e–f, 8a, b and Supplementary Fig. 1c, e, f).** Fixed primary rat cortical neurons were acquired using an SP8 confocal microscope or an ImageXpress Micro 4 High Content Screening System. In Fig. 2e and Supplementary Fig. 1e, f, β-III-tubulin (TexasRed filter set: 562 ± 20/624 ± 20 nm, dichroic filter 593 nm) was imaged using the ImageXpress Micro 4 High Content Screening System at 500 ms with a 20× air immersion lens (Ph1 S Plan Fluor ELWD ADM; NA 0.45). For fluorophore excitation, the in-built Lumencor Sola Light (SE 5-LCR-QB) was set to 100 "lumencor intensity" units in MetaXpress (v6.2.2.) software. This same set-up was used in Supplementary Fig. 1c for Hoechst (DAPI filter set: 377 ± 25/447 ± 30 nm, dichroic filter 409 nm) with a 204 ms exposure time and Fig. 8b for β-III-tubulin (FITC filter set: 475 ± 17/536 ± 20 nm, dichroic filter 506 nm) and Hoechst (DAPI filter set) with 100 ms and 204 ms exposure times respectively. For Fig. 8a, b, Synapsin I, MAP2 and Hoescht were imaged using the SP8 confocal microscope with a 63× oil objective (HC PL APO CS2; NA 1.4) and the following settings: 405, 488, and 638 nm lasers with 26.7, 18.3, and 1 respective power intensities. The following detector settings for Synapsin I, MAP2 and Hoescht images were, respectively selected for: 410–483 nm (HyD) at gain 100, 493–643 nm (PMT) at gain 768.8, and 643–781 nm (HyD) at gain 40.3, Pinhole 1 AU, zoom 1×, and 400 Hz scan speed.

**Imaging fixed hPSC-derived neurons (Fig. 8a, b).** Images of fixed hPSC-derived neurons were acquired using a Leica SP8 confocal microscope or ImageXpress Micro 4 High Content Screening System. MAP2 (Cy5 filter set: 628 ± 20/692 ± 20, dichroic filter: 660 nm) and Hoechst (DAPI filter set) from Fig. 7b were both imaged using the ImageXpress Micro 4 High Content Screening System with a respective 50 and 204 ms exposure time using a 20× air immersion lens (Ph1 S Plan Fluor ELWD ADM; NA 0.45). For all image acquisition using the ImageXpress, the in-built Lumencor Sola Light (SE 5-LCR-QB) was set to 100 "lumencor intensity" units in MetaXpress (v6.2.2.) software. In Fig. 7a, Synapsin I, MAP2 and Hoechst were imaged using a Leica SP8 confocal microscope with a 63× oil objective (HC PL APO CS2; NA 1.4) and the following settings: 405, 488, and 638 nm lasers at 1%, detectors 410–483 nm (HyD) at gain 142, 493–643 nm (PMT) at gain 768.8, and 643–781 nm (HyD) at gain 32.5, Pinhole 1 AU, Zoom 1×, Scan speed 400 Hz.

**Imaging live primary rat cortical neurons (Figs. 2e, 3a–c and Supplementary Figs. 1d, f, 4b, 5, 8a)**. Live rat cortical neurons labeled with Neurofluor™ NeuO (FITC filter set: 475 ± 17/536 ± 20 nm, dichroic filter 506 nm) and Hoechst (DAPI filter set: 377 ± 25/447 ± 30 nm, dichroic filter 409 nm) were imaged using an ImageXpress Micro 4 High Content Screening System set at 25 and 204 ms exposure times respectively with a 20× air immersion lens (Ph1 S Plan Fluor ELWD ADM; NA 0.45). NeuroFluor™ NeuO has an excitation/emission spectrum of 470/555 nm. The in-built Lumencor Sola Light (SE 5-LCR-QB) was set to 100 "lumencor intensity" units in MetaXpress (v6.2.2) software for fluorophore excitation. Phase-contrast images (Fig. 3a–c and Supplementary Figs. 4b, 5, 7a) of live rat cortical neurons were taken with an Olympus CKX53 microscope with an SC100 digital camera and a 10× dry objective (CACHN10XIPC; NA 0.25).

**Imaging live fused dorsal and ventral forebrain organoids (Fig. 2d)**. Fused dorsal and ventral forebrain organoids were imaged using a Leica SP8 confocal microscope with 10× dry objective (HC PL APO CS2, NA 0.4) and the following settings: 488 nm laser at 5% power intensity, detector 493–547 nm (HyD) at gain 176.8, Pinhole 1 AU, Zoom 1×, Scan speed 600 Hz.

**Imaging live hPSC-derived neurons (Fig. 2a–c and Supplementary Figs. 2a–b, 3, 8a)**. Fluorescent images of live hPSC-derived neurons transfected with green-fluorescent-protein (GFP) lentivector were imaged at 200 ms exposure time using a PCO.Panda 4.2 (16-bit) digital camera and Olympus BX51 upright microscope with 40× water immersion lens (LUMPLFLN40XW, 0.8NA, Semiapochromat). A cool-LED pE300 illumination unit was set to blue (460 nm) at 1% power (0.1 mW) and light passed through a single band FITC/Cy2 filter (Cat. No. 49002, Chroma) with a 470 nm excitation (±40 nm) and 525 nm emission (±50 nm) wavelength. Single neurons were perfused with BrainPhys™ (Cat. No. 05790, STEMCELL Technologies) basal media bubbled with a mixture of $CO_2$ (5%) and $O_2$ (95%) and maintained at room temperature. Perfusates were then switched to BrainPhys™ Imaging (Cat. No. 05796, STEMCELL Technologies) and then artificial cerebrospinal fluid (ACSF). Imaging parameters remained constant across all live neuron images in each perfusate. Phase-contrast images (Fig. 7a) of live hPSC-derived neurons were taken with an Olympus CKX53 microscope with an SC100 digital camera and a 10× dry objective (CACHN10XIPC, NA 0.25).

**Quantification of cell viability with imaging for phototoxicity experiments**. Live primary rat cortical neurons (Figs. 3d and 8b): neuron cultures for quantification were set up in triplicate wells and fixed for immunostaining with β-III-tubulin. Twenty-five images per well were taken using an ImageXpress Micro 4 High Content Screening System (Molecular Devices) for a total of 75 images per condition. Using a custom module designed for neuron quantification in MetaXpress 6.2.2 software (Molecular Devices), neurons in each image were counted based on the following parameters. Nuclei are masked if between 10.5 and 35 μm in width and 90 and 350 μm² in area with an intensity above the local background of 500. Objects with a max intensity standard deviation above 2000 were excluded (excludes apoptotic nuclei). Masked area is then queried for presence of β-III-tubulin positive staining with an average intensity above 400. Images, where the neuron count was above 55, were excluded due to increased error in the presence of high nuclei counts. The average number of neurons per cm² was calculated based on the image size (0.00508 cm²) produced from the 20× objective. Fixed hPSC-derived Neurons (Fig. 8b): Neurons for quantification were set up in duplicate wells and fixed for immunostaining on days 14–21. Sixteen images were taken per well and quantified as described above based on the presence of MAP2 positive staining. The average number of neurons per cm² was calculated as described above.

**Fluorescent signal-to-background ratios**. iPSC-derived human neurons (Fig. 2b, c): the mean "signal" intensity of human neurons expressing synapsin:GFP were analyzed per field-of-view (FOV) by computing the mean gray values of ROI manually selected around soma and/or neurites with ImageJ's "multimeasure" function[57]. Whether the signal intensity at soma and neurites were analyzed together or individually is detailed in figure legends. To analyze the mean 'background' intensity in each test media, ROIs were traced around regions without neurons (i.e., cellular bodies or neurite projections) and the mean gray values generated. Signal-to-background ratios were calculated using the following formula: ((signal intensity—background intensity)/background intensity). Per FOV, "signal" and "background" analyses were repeated using the same ROIs across test conditions. Primary rat cortical neurons (Fig. 2f and Supplementary Fig. 1c–e): mean "signal" intensity analysis on the fluorescence staining of fixed and live primary rat cortical neurons was conducted per FOV using ImageJ's[57] "3D object counter" (Fig. 2f) or "Triangle" auto-threshold (Supplementary Fig. 1c–e) in mean gray value units. In Fig. 2f, the fluorescence "signal" from Hoechst, NeuO and β-III-tubulin above 3600, 600, and 7000 (mean gray value units) were masked, respectively. Mean "signal" intensity values of masked regions greater than 30 pixels in size (pixels²) were computed with "3D object counter". In Supplementary Fig. 1c–e, ROIs of Hoechst, NeuO and β-III-tubulin fluorescence stains were generated from binary masks obtained from autoselection/thresholding. ROI outlines were overlapped onto the corresponding original image(s). ImageJ's "multimeasure" function was used to calculate the mean "signal" intensity of masked

regions greater than 30 pixels in size (pixels²). For mean "background" intensity analysis, "background" ROIs (i.e., with no cellular bodies or neurite projections) were generated per FOV from binary masks obtained using ImageJ's "Huang dark" threshold (Fig. 2f) or "Triangle dark" auto-threshold (Supplementary Fig. 1c–e). "Background" ROIs were overlapped onto the corresponding original image(s) and the mean gray value units computed per FOV using ImageJ's "multimeasure" function[57]. Threshold settings remained constant for each fluorescent stain to maintain consistency between replicates. Objects on edges were excluded from all mean intensity results. Signal-to-background ratios were calculated using the following formula: ((signal intensity – background intensity)/background intensity).

**Image analysis per pixel (Fig. 2b and Supplementary Fig. 2b)**. GFP intensity values were computed per pixel from ROIs selected at "background", "neurites", and "soma" regions, as described under "Fluorescent Signal-to-Background Ratios" for iPSC-derived neurons. Using Image J's "Histogram" command[57], ROIs were analyzed using 1240 bins in the intensity range from 60 to 1300. Mean GFP intensity results across all FOVs were graphed either against the cumulative percentage of pixels (Fig. 2b) or the number of pixels (Supplementary Fig. 2b). All data were paired across test conditions. The number of photo-counts of the camera sensor without cells and media were included and labeled as "dark" (Fig. 2b and Supplementary Fig. 2b).

**LED exposure for phototoxicity experiments (Fig. 3a–d and Supplementary Fig. 5)**. A string of battery-operated blue (Cat. No. ED0193L, ER CHEN), red (Cat. No. ED00021, ER CHEN) and violet (Cat. No. UV443528WP, AMARS) LED lights were wrapped around a tissue culture plate. The plate containing the lights was then placed above the test culture plate (containing cultures of primary E18 rat cortical neurons) in a temperature-controlled incubator. The test plate was exposed to light for 12 h total (two bouts of 6 h, 24 h apart) for blue LEDs (450–475 nm; 14 Lux ± 1.41 S.E.M), 18 h for red LEDs (620–740 nm; 55 Lux ± 4.99 S.E.M) and 6 h for violet LEDs (395–405 nm; 430 Lux ± 3.70 S.E.M). Light meter readings were taken of the plate containing the LEDs at eight points over the plate area with a Sper Scientific™ model 840020 light meter (Cat. No. 840020, Sper Scientific).

**Mitochondrial membrane potential and cell viability assessment (Fig. 3k and Supplementary Fig. 4c)**. Primary rat cortical neurons were cultured in 96 well plates precoated with 30 μg/ml poly-D-lysine (Cat. No. P7280, Sigma) at 23,400 cells/cm². On day 13 or 14, cultures were exposed to violet LED light for 1 h in a humidified incubator at 37 °C with 5% $CO_2$ and 21% $O_2$ with a rest period of 2–3 h post-exposure before fluorescent assays. For $H_2O_2$ exposure, 3% (0.88 M) hydrogen peroxide solution (Cat. No. 88597, Sigma) was diluted to 100 μM final concentration in culture wells and cells were incubated for 4 hours in a humidified incubator at 37 °C with 5% $CO_2$ and 21% $O_2$. For fluorescent assays, the culture medium was removed from all wells and replaced with 200 μl BPI. Twenty microliter of CellTiter-Blue® Reagent (Cat. No. G8081, Promega) was added per well in triplicate wells for each medium condition. After incubation in 37 °C with 5% $CO_2$ and 21% $O_2$ for 30 min, a JC-1 probe reconstituted in dimethyl sulfoxide (Cat. No. T3168, Thermo Fisher Scientific) was added to the second set of triplicate wells to 2 μg/ml final concentration. Plates were then incubated for another 30 min at 37 °C with 5% $CO_2$ and 21% $O_2$. Wells containing JC-1 probe were subsequently washed by performing two half-medium changes with BPI. Fluorescence was read using a SpectraMax M5 Multimode Microplate Reader (Molecular Devices) at wavelengths: 485 nm Ex/535 nm Em (JC-1 monomers), 535 nm Ex/595 nm Em (JC-1 aggregates), and 560 nm Ex/590 nm Em (CellTiter-Blue® cell viability). Fluorescence readings were taken in triplicate for each well in a horizontal scan pattern. Blank values were subtracted from all readings in SoftMax Pro software (version 7.1). Values were averaged across triplicate readings and wells for each condition. Ratio of JC-1 aggregates to monomers was calculated by dividing the value of 595 nm emission by the value of 535 nm emission.

**Media cytotoxicity comparisons (Fig. 3e, f)**. LDH assay for cell viability: LDH release in human neuronal cultures treated with light-stimulated neuronal maturation media was measured to assess loss of cell viability. Media was made and stimulated with ambient tissue culture hood light for 24 h as described under "$H_2O_2$ measurements". Human midbrain and cortical neurons were matured and treated in 48-well (Fig. 3g) and 96-well (Fig. 3h) culture plates, respectively. Treatment: neuronal maturation media was replaced with 24-h light stimulated media (Day 0). Half-Media changes were performed every 24 h (Day 1, Day 2…) using fresh 24-h light stimulated media. Supernatant samples were collected on Day 0 immediately after feeding, and immediately before feeding on Day 1 onwards for the LDH assay. Samples were frozen at −20 °C prior to LDH measurement. LDH assay: LDH was measured using CytoTox 96® nonradioactive cytotoxicity assay (Cat. No. G1780, Promega) as per the manufacturer's protocol. Formazan absorbance at 490 nm was measured by GloMax microplate reader (Promega) using GloMax (v3.1) software. Blank LDH levels were subtracted from Treatment LDH values. Normalization can be found in figure legends. For Fig. 3e, f, luminescence measurements were normalized first to BPI results for each respective day, then values were renormalized to day 0.

**H₂O₂ measurements in light stimulated media (Fig. 3g, j).** Media: BrainPhys and BPI basal media were supplemented to make neuronal maturation media as described under "Human Pluripotent Stem Cell (hPSC)-derived Neuron Culture" (Method #1). Media was made fresh for each experiment and stimulated with blue or ambient light for 24 h. Lumos optical stimulation (at 475 nm): 500 µl of media was added to the wells of a 48-well Lumos MEA plate (Cat. No. M768-tMEA-48OPT, Axion Biosystems). The 48-well Lumos MEA plate was maintained at 37 °C, 5% CO₂ environment within a Maestro Pro MEA system (Axion Biosystems). The media were stimulated with blue light (475 nm) for 24 h using a Lumos optical stimulator (Axion Biosystems) adapted for 48-well MEA plates. Stimulation was repeated with the following settings: 10 × 5 ms flashes of blue LED light at 10 Hz and a 30-s interburst interval. Incremental blue LED power intensities were used (0, 3, 5, 12, 27, 53, 79, and 105 mW). The light intensity (mW) of blue light emitted from a Lumos optical stimulator (Axion Biosystems) was measured using a PM100D Compact Power and Energy Meter Console (Thor Labs) set to 475 nm (Supplementary Fig. 4a). Ambient light stimulation: 500 µl of media was added to the wells of a 48-well plate (Corning CoStar). Forty-eight-well plates were placed in a Cellgard ES ClassII biological safety cabinet (Cat. No. S480-600E, Nuaire) and stimulated for 24 h with a 58-watt NL-T8 Fluorescent Lamp Spectralux©Plus (Radium) located at the ceiling of the cabinet. The lamp is 1.5 m in length with a 2.5 mg mercury content and medium bi-pin G13 bas, giving rise to a 5200 lm luminous flux. Control plates were wrapped with aluminum foil to protect them from light. H₂O₂ Luminescence Measurements: H₂O₂ measurements were made using ROS-Glo™ (Cat. No. G8820, Promega) kits as per the manufacturer's protocol. Upon concluding light stimulation, H₂O₂ substrate was added to sample media to a final concentration of 25 µM. Samples were incubated for 2 h in a dark humidified incubator at 37 °C with 5% CO₂ and 21% O₂. Samples were transferred to an opaque black-walled 96-well plate in triplicates, combined with an equal volume of ROS-Glo™ Detection solution, and incubated for 20 min before relative luminescence units (RLU) was measured on a GloMax microplate reader (Promega) using GloMax (v3.1) software.

**Light spectra recordings.** Ambient light (Fig. 3h): to assess emission spectra from a 58-watt NL-T8 Fluorescent Lamp Spectralux©Plus (Radium) in a Cellgard ES ClassII biological safety cabinet (Cat. No. S480-600E, Nuaire), a Flame VIS-NIR spectrometer (Ocean Insight) with a 10 µm slit was placed in the cabinet, facing the white-light fluorescent lamp. The emission spectra were captured using an integration time of 2 ms. LUMOS optical stimulator at 475 nm (Fig. 3i): To assess emission spectra from the Lumos optical stimulator (Axion Biosystems) set to a wavelength of 475 nm, a Flame VIS-NIR spectrometer (Ocean Insight) installed with a 10 µm slit was placed approximately 1 cm away from a single LED on the Lumos optical stimulator. The emission spectra were captured using an integration time of 1 ms.

**Calcium imaging.** hPSC-derived neurons attached onto coverslips in 48-well plates were incubated with a 0.5 µl of Fluo4-AM(1 mM; Cat. No. F14201, Thermo Fisher Scientific) in 500 µl of neuromedium, per well (final Fluo-4 AM concentration in media: 1 µM). Incubation occurred for 20 min in a humidified incubator at 37 °C with 5% CO₂ and 21% O₂. Excess dye was removed by washing three times with ACSF. The cells were then transferred into a room-temperature recording chamber continuously perfused with ACSF and bubbled with a mixture of CO₂ (5%) and O₂ (95%) for an additional 30 min at room temperature to allow de-esterification. Fluo-4 AM concentrations lower than 2 µM and loading times less than 30 min did not induce harmful effects to the neurons[58]. Time-lapse image sequences were acquired at 5 Hz over 4 min with a region of 248 × 248 pixels. Images were recorded at 200 ms exposure time using a PCO.Panda 4.2 digital camera and an Olympus BX51 upright microscope with a 10× water immersion lens (UMPLFN10XW, NA0.3, Semi-Plan Apochromat) and FITC/Cy2 filter set (Cat. No. 49002, Chroma). A cool-LED pE300 illumination unit was set to 460 nm (1%; 0.1 mW) to exciteFluo4-AM. Imaging parameters were kept constant when recording in different test media. To assess network activity in response to different media, the following were perfused into the bath over 6 min at a constant rate of 0.32 ml/min: ACSF, BPI, BP, NEUMO, FluoroBrite. At the end of the experiment before discarding the culture 1 µM Tetrodotoxin (TTX; Cat. No. ab120054, Abcam) perfusate was added to the basal media to block voltage-gated sodium channels and action potential generation. The orders of media perfusion were alternated, apart for TTX, which was always perfused last. Images were then processed using ImageJ software. A heat-map LUT was applied to each image sequence to visualize calcium events as changes in fluorescence over baseline. A ROI was placed around each soma displaying a defined cellular morphology and at least one clear neuronal calcium event (dF/F > 5%, fast rise, slower decay). A time series for each ROI was calculated using ImageJ[57] and Microsoft Excel before analysis in Clampfit (v10.7). Different types of calcium events were manually categorized as either calcium waves, spikes or a combination of calcium waves and spikes (Supplementary Fig. 8a).

**Patch-clamping.** Individual coverslips containing neurons were transferred into a recording chamber at room temperature (21 °C) continually perfused at 0.32 ml/min with either ACSF or BrainPhys™ Imaging media bubbled with a mixture of

CO₂ (5%) and O₂ (95%). Whole patch recordings were made on neurons expressing the previously infected synapsin:GFP lentiviral vector. Targeting whole-cell recordings were achieved via a 40× water-immersion objective, and a PCO. Panda 4.2 digital camera on an Olympus BX51 microscope. A cool-LED pE300 illumination unit at 460 nm was used to visualize neurons expressing synapsin: GFP. Patch electrodes were filled with internal solutions containing 130 mM K-gluconate, 6 mM KCl, 4 mM NaCl, 10 mM Na-HEPES, 0.2 mM K-EGTA, 0.3 mM GTP, 2 mM Mg-ATP, 0.2 mM cAMP, 10 mM D-glucose, 0.15% biocytin and 0.06% rhodamine for somatic (open tip resistance 3–5 MΩ) whole-cell recordings. The pH and osmolarity of the internal solution were close to physiological conditions (pH 7.3, 290–300 mOsmol). AMPA-receptor mediated events were observed exclusively in voltage-clamp at −70 mV (close to anion reversal potential). GABAa-receptor-mediated kinetics were observed exclusively in voltage-clamp at 0 mV (close to cation reversal potential). Recorded membrane potential values were adjusted for pipette offset. Electrode capacitance was compensated for in cell-attached mode. Signals were low-pass filtered (DC to 20 kHz) and acquired by PClamp software. Whole-cell recordings were amplified via a Digidata 1440A/Multiclamp series 700B. On average the neurons recorded had an access resistance of ~22 ± 7 MΩ (Mean ± Stdev). The average patch-pipette resistance used was 3.8 ± 0.5 (Mean ± Stdev). Final data were analyzed using a combination of Clampfit (v10.7) and GraphPad Prism 8 and were corrected for liquid junction potentials (10 mV).

**Action potential (AP) type classification.** AP type classification was based on previously published definition (Bardy et al.[28]) as follows: "Type 0 cells" are most likely non-neuronal and do not express voltage-dependent sodium currents. "Type 1 neurons" express small Nav currents but are not able to fire APs above −10 mV. The limit of −10 mV was chosen as it is close to the reversal potential of cations (0 mV), and a sign of healthy mature APs. "Type 2 neurons" fire only one AP above −10 mV, which is typically followed by a plateau. "Type 3 neurons" also fire an AP above −10 mV and one or a few aborted spikes below −10 mV. "Type 4 neurons" fire more than one AP above −10 mV but at a frequency below 10 Hz. "Type 5 neurons" fire APs above −10 mV at 10 Hz or more.

**Optogenetics during patch-clamping.** Blue light stimuli (475 nm) were controlled by a computer-controlled trigger signal to the LED illuminator (pE300; CoolLED) synchronized with the pClamp protocols through a Master 9 programmable pulse stimulator (A.M.P.I.). The patched neurons were stimulated with a burst consisting of ten flashes of blue (475 nm) light of a duration of 5 ms at 10 Hz. Each burst stimulus was repeated at least ten times every 10 s for each cell. The intensity of the blue light at 475 nm tested here ranged between 0.1, 0.2, and 0.4 mW. Illumination was achieved via an Olympus BX51 upright microscope, through a 40× water immersion lens (LUMPLFLN40XW, 0.8NA, Semiapochromat) focused on the soma and surrounding of the patch-clamped neurons.

**Multielectrode arrays (MEAs).** MEAs recordings were performed in two independent laboratories with slightly different equipment and protocols. The two methods differed as follow:

**MEA method #1 (Figs. 5 and 7f–i).** Cell culture on MEA plates #1: The neuronal cultures were generated from NPCs in the neural maturation medium (NMM) as outlined under hPSC-derived neuron culture (Method #1). Fourteen days after neuronal maturation in NMM, cultures were dissociated using Accutase (Cat. No. 07920, STEMCELL Technologies) and resuspended in NMM at 15,000 viable neurons/µl. A 10 µl droplet of cell suspension was added directly over the recording electrodes of each well of the MEA, which were previously coated with 10 µg/ml poly-L-ornithine (Cat. No. P3655, Sigma) and 5 µg/ml laminin (Cat. No. 23017015, Thermo Fisher Scientific). Sterile water was added to the area surrounding the wells, and the MEA was incubated at 37 °C, 5% CO₂ in a cell culture incubator. After 1 h, 300 µl NMM was gently added to each well. Half of the culture medium was exchanged with fresh medium three times per week. Fifty-two days after maturation commenced, the medium was replaced with one of three different media: BrainPhys™ Imaging, artificial cerebrospinal fluid (ACSF) or BrainPhys™, each supplemented with the midbrain or cortical basal media supplements. MEA recording and analysis #1: MEA recordings were taken by a Maestro Pro MEA system (Axion Biosystems). Lumos MEA 48 plates (Cat. No. M768-tMEA-48OPT, Axion Biosystems) were maintained at 37 °C with 5% CO₂ environment during recordings. Recordings were not made for at least 10 min after transferring a plate into the MEA. Version 2.4 AxIS acquisition software (Axion Biosystems) was used to sample voltage potentials simultaneously across 16 electrodes per well. The sampling frequency was 12.5 kHz. The threshold for detecting spikes was set on a per-electrode basis and was defined as the voltage exceeding six standard deviations away from the mean background noise..spk files were analyzed using NeuralMetricTool version 2.5.1 software (Axion Biosystems). An electrode was defined as active with a mean firing rate greater than 0.017 Hz (>1 spike per min). Optogenetics on MEA #1: Optical stimulation parameters were configured using Stimulation Studio software in AxIS. Optical stimulation was performed by a Lumos optical stimulator (Axion Biosystems). The Lumos stimulation protocol was as follows: Every 20 s, wells were simultaneously exposed to a stimulation burst

consisting of ten flashes of blue (475 nm) light, each lasting 10 ms with a 100 ms interval between each flash. Lumos blue LED stimulation at 100% intensity is >3.9 mW/mm[2] according to the manufacturers. The intensities of stimulation used here ranged from 5 to 50%. Spontaneous and light-evoked firing rates were recorded for 10 min postfeeding neuronal cultures.

**MEA method #2 (Fig. 8c).** Cell culture on MEA plates #2: Frozen neural progenitor cells (NPCs) induced pluripotent (iPS) stem cells (characterized in the ref. [51]) were differentiated into neuronal precursors with the STEMdiff™ Neuron Differentiation Kit (Cat. No. 08500, STEMCELL Technologies). Neuronal precursor cells were then frozen with Cryostor10 (Cat. No. 07930, STEMCELL Technologies) in liquid nitrogen storage and thawed later for experiments. Post-thaw, cells were plated at 10,000 cells in a 40 µl droplet of STEMdiff™ Neuron Differentiation medium in a Cytoview MEA 48 plate (Cat. No. M768-tMEA-48B Axion Biosystems) pre-coated with 15 µg/ml poly-L-ornithine (Cat. No. P4957, Sigma) and 10 µg/ml laminin (Cat. No. L2020, Sigma) (12-wells total). After a 2-h incubation in a temperature-controlled incubator, 150 µl of the medium was added to each well. On post-thaw day 1, 150 µl of complete BP was added to each well. One well of the total 12 was eliminated from testing due to poor cell adhesion. Cells were cultured for 6 weeks in complete BP with half-media changes performed every 3–4 days. Subsequently, media in 5 out of 11 of the wells was exchanged with complete BPI for a period of 2 weeks before switching back to BP. MEA recording and analysis #2: Spontaneous neuronal activity was acquired at 37 °C under a 5% $CO_2$ atmosphere using a Maestro system (Axion Biosystems) at a sampling rate of 12.5 kHz/channel. For recording, a Butterworth band-pass filter (200–3000 Hz) was applied and the adaptive threshold spike detector was set at 6× standard deviation. A 15-min recording was taken twice a week. Data from the last 10 min of each recording was exported for analysis using AxIS software v2.5.1 (Axion Biosystems). An electrode was defined as active with a mean firing rate greater than 0.017 Hz.

**Statistics.** Statistical analyses were performed with Graphpad Prism 8. Significance was assessed between different media and BrainPhys™ Imaging groups using the following statistical tests: two-tailed nonparametric unpaired (Mann–Whitney) test, two-tailed unpaired *t*-test; two-tailed nonparametric paired (Wilcoxon) test; and one-sided sum-of-squares *F*-test.

**Ethics.** Experimental procedures with human cells were performed in accordance with The Women's and Children's Health Network Human Research Ethics Committee in South Australia (HREC/17/WCHN/70). Human cells were collected from patients with informed consent. H1 (WA01) or H9 (WA09) human embryonic stem cells were obtained from WiCell (Agreement No. 20-W0500). Ethics approvals for tissue collection and sale are retained by the vendor. Rodent tissue was purchased from a commercial supplier (Brain Bits, LLC) whose procedures are approved by an Institutional Animal Use and Care Committee.

**Reporting summary.** Further information on research design is available in the Nature Research Reporting Summary linked to this article.

## Data availability
The data that support the findings of this study are available from the corresponding author upon reasonable request.

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

## Acknowledgements

We thank Mr Sebastian Loskarn and Dr Zarina Greenberg for technical assistance. This work was generously supported by Perpetual Impact Philanthropy, the Brain Foundation, Rebecca L. Cooper Foundation, Ian Potter Foundation, Boileau Corporate Philanthropy, Parkinson's South Australia, MRFF Australian Government and Department of Health, Australian Research Council LIEF grant (to C.B.); Centre for Nanoscale and Biophotonics (CE140100003), Future fellowship (FT180100565) (to M.R.H.), the US National Research Council Senior Fellow Program to the US Air Force Research Laboratory (to A.M.J.), the Netherlands Organisation for Scientific Research Rubicon Fellowship 019.163LW.032 (to M.vdH.), and the RMIT University Vice-Chancellor's Research Fellowship (to P.R.) and The Australian Government Research Training Program Scholarship (to M.Z., B.M., and R.A.). Flow cytometry analysis and cell sorting were performed at the South Australian Health Medical Research Institute (SAHMRI) in the ACRF Cellular Imaging and Cytometry Core Facility. The Facility is generously supported by the Detmold Hoopman Group, Australian Cancer Research Foundation and Australian Government through the Zero Childhood Cancer Program.

## Author contributions

C.B., C.M., V.M.L., and E.K. designed the project. K.M., B.M., R.A. and M.Z. executed and analyzed the fluorescent phototoxicity experiments. M.Z., K.M., A.A., L.H.C., A.J.H., J.T.H. and A.M. performed the absorbance and fluorescence assays. B.M., R.A., C.M., M.Z., and K.M. performed the MEA experiments. K.M., C.M., B.M., J.T.R., and M.vdH. performed tissue culture. M.Z., and A.P.S. executed and analyzed the patch-clamping, calcium imaging, and optogenetics experiments. A.M.J. and P.R. performed the absorbance analysis. C.B. wrote the manuscript with contributions from M.Z., K.M., and M.vdH., and feedback from all authors.

## Competing interests

C.B. is the inventor on the BrainPhys patent. STEMCELL Technologies is distributing the BrainPhys media. K.M., C.M., A.A., L.C., A.H., and E.K. are employees of STEMCELL Technologies. All other authors declare no competing interests.
