## [Peer Review File · Nature Communications]

Reviewers' Comments:

Reviewer #1:

Remarks to the Author:

To improve the quality of the optical experiments, Zabolocki et. al. developed a new medium, BrainPhys Imaging (BPI) that is based on the formulation of the original BrainPhys medium. Authors prove that this BPI is superior to other mediums used so far through the morphological, electrophysiological and optical analyses (optogenetic and calcium imaging experiments). And also, it found that the BPI is free of cell phototoxicity occurred in the other medium and consequently emphasizes the availability of the entire light spectrum, allowing for unique optical studies.

The results show the importance and excellence of the BPI at the same time. This paper is a very important technological development for experiments with various types of human neurons in the future.

Overall, the story is very well organized and includes convincing contents. It is an important result, and this manuscript was reasonably well written. However, there are still a few allegations to fully understand the role of BPI for functional analyses of neuronal culture.

Main Issues

1. In Figure 1, the authors measured the osmolarities of different mediums in absence of neurons. Of course, it is an important factor explaining the condition of medium. However, the changes in osmolality with neuronal growing is also a critical point to understand the role of medium.
2. In Figure 3, the passive and active membrane properties are important data to determine the identity of neurons that are distinct from other cell types. Here, the authors should specifically analyze the properties of membrane according to the growth of neurons in order to clarify the effect of the medium, standard BP+ supplements medium, on the growth of neurons. Through these data theorem, it is possible to detect the timing of neuronal maturation. Therefore, the author needs specific data on how the membrane properties appear as neurons grow.
3. In neuronal network, synapses are the most basic unit of neuronal communication, and especially, in chemical synapses, the synaptogenesis is the most important measure for the functional neurons. Therefore, authors consider how/when synapse formation was occurred.
4. In figure 3, the amplitudes and frequencies of spontaneous EPSC measured from BPI medium is more widely distributed than the values measured from ACSF. It would be nice to describe/express the author's opinion in DISCUSSION part.
5. It has been known through several experiments that a calcium store exists in rat neurons, which also affect calcium spike. In figure 4, it can be seen that calcium spike in human neurons rarely occurs in the presence of TTX, Did authors consider the effect of calcium stores on calcium spikes in this human neurons?

Minor Issues

1. It is not clear when the lenti virus is infected/transfected with neurons and when the experiments were performed after infection/transfection.

Reviewer #2:

Remarks to the Author:

In this paper the authors describe the development of a new neuronal culture media designed for optimal live-cell imaging of neurons in vitro. This media is an adapted formulation of the BrainPhys media first reported by Bardy et al in 2015. In this paper, the authors adjust the formulation of this medium to reduce phototoxicity and autofluorescence, while still supporting electrophysiological activity of the cultured neurons. As the use of iPSC-derived neuronal cultures and organoids are becoming widely adopted, the development of media that support physiological neuronal activity and allow live-cell (imaging) studies – such as the BPI media described here - is

highly relevant and will be of broad interest to the scientific community and the Nature Communications readership.

In general, the paper is well written, and the results support the claims. There are a few sections where the paper could be improved or the data could be strengthened, mostly through improving the analysis or data presentation. There are several sections of the methods that need to have significantly more information to allow others to reproduce the experiments.

Major comments

1. In Figure 1B, it is impossible to judge the values for the lower autofluorescence media. Please make separate curves for that. The data can all be relative to PBS for comparison but the low intensity autofluorescence needs to be visible for the reader to judge the significance of the differences.
2. In Figure 1D-F, the autofluorescence of the media is significantly higher for the BP media relative to the BPI. However, in Figure 1G and 1I the values for the different media are surprisingly similar. This discrepancy needs to be explained.
3. The quantification of fluorescence background in Figure 1G doesn't seem to reflect the images that are shown. Are the intensity display settings the same for all the panels in Figure 1G? If not explain why and how they were adjusted.
4. It is unclear if the background intensity is subtracted before measuring the signal. It is important to do that, or it will severely impact the S/B ratios for the different wavelengths. If the signal intensity was background corrected, then the differences in signal to background ratio seem to come primarily from an increase in signal, rather than a reduction of the background, in contrast to what is stated on page 5, line 140. If it was not the data needs to be recalculated.
5. Many details are missing in the methods section. See minor comments below for details. The way the methods are now the imaging and image analysis could not be reproduced.
6. Were antibodies validated?
7. Fig 1I: the results for ACSF are not mentioned in the results section, they should be discussed.
8. In Figure 2, a quantification of live neurons pre-light should be shown in B, D and F to establish a baseline. The results could be presented as a change in cell number.
9. Experiments showing photobleaching of the different dyes in the different media would give an idea if dye photobleaching is contributing to phototoxicity.
10. Experiments showing endpoint intensity of a ROS probe or mitochondrial morphology would help demonstrate phototoxicity with the different media. This may pick up more subtle phototoxic effects that do not show up as cell death for example.
11. There is a statement in the methods section that contrast and exposure settings were adjusted between timelapse imaging sequences when necessary. Does that mean the light exposure of the samples differs for different media tested? If this is the case this must be carefully incorporated into the results to make sure differences in intensity are not playing a role in the observed phenotypes.

Minor comments

1. Avoid using terms like better and worse be specific. For example, on lines 129 and 139 just say autofluorescence is lower not better. Similarly, the Fig. 1 title should be signal-to-background ratio is higher... not improved.
2. All cell images should be in greyscale unless they are part of a colour overlay.
3. In Figure 1G and 1I, use a gamma factor to show neuronal protrusions and autofluorescence without saturating the cell body. Put these image display properties in the figure caption so readers know it is a non-linear display, but this will avoid the saturated cell bodies in the images.
4. Plots in Figure 1G should go to 0. Add a second scale break on the y-axis if necessary.
5. In Figure 1B, the combination of colors and thickness of the lines, makes some curves barely visible and the plots hard to read.
6. In depth details of the following are missing:
 - a. Imaging parameters for the ImageXpress Micro 4 HCS
 - b. Imaging parameters for the SP8
 - c. Image analysis settings for the custom module designed for neuron quantification in MetaXpress

6.2.2

d. Imaging parameter for the Olympus IX73 – e.g. what are the properties of the 40x lens? Lens type, NA, camera exposure time, light source?

7. In the methods section, please provide the properties of the objectives used (not only magnification, but also NA and color correction (Neofluor, apochromat, plan-apochromat?))

8. In Figures 3-5, why wasn't BP media included as a control?

9. More information on the autofluorescence media measurements with the BMG Labtech is needed. What is the bandwidth of the filters, what light source was used, intensity of light, exposure time, what detector, etc?

10. Secondary antibody catalogue numbers, lot numbers and concentrations are missing.

11. Catalog and lot numbers (if appropriate) should be added for all reagents and tools mentioned in the Methods section.

12. This article should be cited. "Assessing phototoxicity in live fluorescence imaging, P Philippe Laissue, Rana A Alghamdi, Pavel Tomancak, Emmanuel G Reynaud & Hari Shroff, Nature Methods volume 14, pages657–661(2017).

13. ImageJ and/or FIJI papers and plugins should be properly referenced.

14. Were any core facilities used for this work that should be acknowledged?

Reviewed by Claire M. Brown, McGill University with assistance from Michiel Krols, Post Doctoral Fellow, McGill University as a review training exercise.

Reviewer #3:

Remarks to the Author:

Significant advances have been recently made in two major in vitro technologies: the synthetic control of cell fate, and the ability to optically measure and manipulate activity in these cells. However, the artificial cellular environment or media that is used to maintain and functionally monitor these cells - particularly neurons - has not seen a parallel advance. This has resulted in unwanted side-effects such as phototoxicity in long-term cell culture experiments, inability to faithfully capture cell-physiological changes, and a concomitant reduction in cell viability. In the study by Zabolocki et al., the authors improve upon their previous formulation (BrightPhys or BP) to generate a new cellular media (BrightPhys Imaging or BPI) which they claim is better for measuring and manipulating in vitro neuronal activity, especially when studying human stem cells in culture.

The authors perform a series of experiments to test fluorescence performance in fixed tissue, and then test their media in a wide range of in vitro applications, including Ca²⁺ imaging and optogenetics. Based on their data, they make the claim that BPI is a superior media formulation compared to other commercially available options. While the updated formulation (BPI) is indeed better in terms of signal-to-background in some applications and ranges of excitation/emission, it is really not demonstrated that this new formulation is a significant advance over their previous one (BP), especially considering that the range and size of overall benefit do not seem that much higher. In addition, the fact that they can generally capture and manipulate physiology in human neuronal cultures has already been previously shown by the same authors.

The major general issues with the paper are: 1) The whole paper is only a very incremental step forward compared to their previous study – and not a significant conceptual advance that would deserve publication in a high impact journal; 2) The study is mostly descriptive, lacking mechanistic data as to why BPI seems to perform better in some aspects compared to its predecessor and other commercially available media, and 3) The paper is purely methodological and as a method is not particularly novel. Since the corresponding author has already published a paper on the BP formulation previously and has covered very similar ground in that paper, the novelty of this report is significantly reduced.

Specific weaknesses

1. The signal-to-background analysis, on which one of the main claims is based, is relatively

simplistic. For example:

a. How does the signal-to-background ratio scale with the actual level of GFP? In Fig. 1, I, the authors have quantified mean GFP intensity in the cell body and dendrites together. Since there is a larger amount of GFP in the cell body, this region seems saturated. This potentially obscures the true signal-to-background ratio. The authors must compare quantities based on a reasonable dynamic range, and avoid saturating pixels.

b. How does the signal-to-background ratio compare across media in Ca²⁺ imaging experiments?
2. The claim that BPI does not result in major phototoxic cell death compared to other media is a very significant claim. However, other than a very surface level exploration of photo-toxicity, there is a lack of explanation as to why BPI does not lead to a significant loss in cell density compared to other media.

a. In Fig.2, the authors compare the effect of light exposure on the viability of primary rat cortical neurons in different media. However, this is a very surface level analysis. The authors must show further details. For example, how temporally dynamic is the loss of cells? Since light excitation/exposure conditions vary widely across experiments, how does the level of exposure scale with loss of cells?

b. What is the mechanism of protection that BPI confers on these cultures? Are there differences in gene/mRNA/protein expression following light exposure across media that provide an explanatory mechanism? Did the authors measure reactive oxygen species and other cytotoxic components in the media post-exposure? At what level does the BPI prevent major cell death?

c. In Fig. 2B-F, the authors compare cell counts post-exposure. Instead, the authors should compare percentage loss in cells pre- and post-exposure.

3. Comparison of BPI to BP physiology experiments – did the authors compare BP to BPI in the electrophysiology, Ca²⁺ imaging, and optogenetics experiments? If BP is comparable to BPI in terms of physiology, then it is a point against the significance of the novelty of BPI compared to BP.

4. With this manuscript, the authors make only a small incremental jump from their previous publication. Being a person who images in cultures is an excellent neural resource; however, several things reduce my enthusiasm for this manuscript. I think the authors do an outstanding job in Figure 1, demonstrating the difference between BP and BPI in terms of signal to noise. In Figure 2, I believe the authors continue to provide convincing data showing the difference between BP and BPI to Photo-toxicity via violet, blue, and red light. However, the major issues for this manuscript start with figure 3. It would have been beneficial for the authors to continue to make a direct comparison between BP, BPI, ASCF throughout Figure 3 and others.

5. In Figure 4, I was surprised the authors used Fluo-4AM to test the Ca²⁺ signaling in the human neuronal cultures. For example, the authors could be missing some of the Ca²⁺ signals due to the harmful effects of Fluo-4AM on neuronal properties via disruption of the Na, K ATPase. I recommend the authors review the downside using Fluo-4AM and such indicators as in Smith et al., 2018 PMID: 29382785. It would have been beneficial for the authors to use viral GCaMP to assessed Ca²⁺ signaling in human neurons. The authors also failed to include BP in the comparison between BPI and ASCF. This was another lost opportunity for the authors to demonstrate why BPI is superior to BP.

6. In Figure 5, the authors missed another opportunity to compare BP, along with BPI and ACSF.

REVIEWER COMMENTS

To all reviewers:

We are extremely grateful for the insightful and constructive feedback from all the reviewers, and for giving us the opportunity to revise on manuscript and address the comments. Addressing all the suggestions enabled us to significantly improve this manuscript, which we hope will be valuable to the scientific community. We also hope you will now find this version suitable for publication.

To address the main suggestions, we have conducted a series of additional experiments, including new osmolality measurements of conditioned media (**Supp Fig 1A**), new media fluorescence measurements (**Figure 1C**), new imaging analysis (**Figure 2**), new photo-toxicity experiments (**Figure 3E-K**), and new patch-clamp and MEA experiments comparing the electrophysiology of human iPSC-derived neurons (**Figure 4Q-V**, **Supp Fig 7**). We have updated the details of all the reagents used in the **supplementary resource table**. **The changes made throughout the manuscript to address the reviewers' suggestions are highlighted in red in uploaded revised version.** Please find also **below the point-by-point responses (blue)** to the comments and suggestions.

Reviewer #1 (Remarks to the Author):

To improve the quality of the optical experiments, Zabolocki et. al. developed a new medium, BrainPhys Imaging (BPI) that is based on the formulation of the original BrainPhys medium. Authors prove that this BPI is superior to other mediums used so far through the morphological, electrophysiological and optical analyses (optogenetic and calcium imaging experiments). And also, it found that the BPI is free of cell phototoxicity occurred in the other medium and consequently emphasizes the availability of the entire light spectrum, allowing for unique optical studies. The results show the importance and excellence of the BPI at the same time. This paper is a very important technological development for experiments with various types of human neurons in the future. Overall, the story is very well organized and includes convincing contents. It is an important result, and this manuscript was reasonably well written. However, there are still a few allegations to fully understand the role of BPI for functional analyses of neuronal culture.

We thank the referee for the positive evaluation of our work and encouraging comments.

Main Issues

1.1. In Figure 1, the authors measured the osmolarities of different mediums in absence of neurons. Of course, it is an important factor explaining the condition of the medium. However, the changes in osmolality with neuronal growing is also a critical point to understand the role of medium.

To address this suggestion, we have measured the osmolality of supernatant collected from human cortical neuronal cultures matured in BrainPhys Imaging (BPI), BrainPhys (BP), BrightCell NEUMO, FluoroBrite, Neurobasal and DMEM/F12 over 14 days. No significant changes in osmolality were found across the 14-day period, *in-vitro*. We added the results in a new Figure S1A and updated the results line (see page 4, line 104-105 of updated manuscript). 'Osmolality' methods have been updated (see page 16, line 555-556 of updated manuscript).

1.2. In Figure 3, the passive and active membrane properties are important data to determine the identity of neurons that are distinct from other cell types. Here, the authors should specifically analyze the properties of membrane according to the growth of neurons in order to clarify the effect of the medium, standard BP+ supplements medium, on the growth of neurons. Through these data theorem, it is possible to detect the timing of neuronal maturation. Therefore, the author needs specific data on how the membrane properties appear as neurons grow. In neuronal network, synapses are the most basic unit of neuronal communication, and especially, in chemical synapses, the synaptogenesis is the most important measure for the functional neurons. Therefore, authors consider how/when synapse formation occurred.

We completely agree that assessing the maturity of neurons and when synapses are formed is an important aspect in the field of hiPSC-derived neurons. We have previously published the advantage of BP for synaptic maturation of hiPSC neurons compared to other media (Bardy et al. 2015; PMID: 25870293). Most importantly, we have also

previously published an in-depth patch-clamping analysis of human neurons maturing in BrainPhys and used PatchSeq to correlate their electrophysiological properties, synaptic formation, morphology and transcriptomic profiles (Bardy et al. 2016; PMID: 27698428). We believe this previous work addresses the reviewer's comment. In the published work, we defined five functional types of neurons (1 to 5), which highly correlate with membrane properties, synaptic function, dendritic complexity and molecular neurodevelopment *in vitro*. Here, we found that BPI did not significantly alter the proportion of these types. New figure S6C showcases the percentage of type 1-5 neurons patch-clamped in either BP or BPI. In comparison to the properties reported in our previous papers, the patch-clamping data reported here are mostly from functionally mature neurons with active synaptic connectivity. This is further demonstrated in Figure 4 which was updated with additional MEA data comparing the synchronicity of neural networks in BP and BPI (See new Figure 4Q-V).

1.4. In figure 3, the amplitudes and frequencies of spontaneous EPSC measured from BPI medium is more widely distributed than the values measured from ACSF. It would be nice to describe/express the author's opinion in the DISCUSSION part.

It is an interesting observation. Even though we have tried to control for the functional maturity of our cultures by only patch-clamping type 5 neurons matured more than 12 weeks *in-vitro* (Figure 3, now Figure 4), amplitudes and frequencies of spontaneous EPSCs remain widely distributed due to inherent neuronal heterogeneity. Based on the properties of previous datasets such variance is quite standard (see Bardy et al 2015, 2016 for example; PMID: 25870293 and PMID: 27698428). We do not know for sure what are the main contributing factors for the EPSC frequencies distribution in this case, and it is not clear to us that the distribution is significantly higher in BPI. However, what we are confident about is that we found no significant differences between the average electrophysiological properties of the cells in BPI, BP and ACSF across multiple readouts.

1.5. It has been known through several experiments that a calcium store exists in rat neurons, which also affect calcium spike. In figure 4, it can be seen that calcium spike in human neurons rarely occurs in the presence of TTX, Did authors consider the effect of calcium stores on calcium spikes in this human neurons?

We appreciate this comment. In Figure 4 (now Figure 5), the influence of calcium stores on calcium spikes in spontaneously active human neuronal cultures was not taken into consideration. Calcium spikes are correlated with action potential generation (Ali and Kwan., 2020; PMID: 31372367) and that calcium store release in rat hippocampal neurons limits action potential generation (Segal., 2018; PMID: 30230988). Specifically, calcium store release increases intracellular calcium and activates calcium-gated potassium currents (Segal., 2018). This increases action potential thresholds and suppresses neuronal activity (Segal., 2018). The importance of elevated calcium concentrations is twofold, as it enables network connectivity by increasing synaptic release while concurrently reducing neuron excitability (Penn et al., 2016; PMID: 26961000). Given that BPI supports optimal electrophysiological activity at both the *network* (new Figures 4Q-V) and *single-cell* levels (Figures 4A-P), we are confident that its formulation is not contributing to significant unphysiological calcium store release and consequently, the suppression of neuronal activity. To further validate this, we have conducted additional patch-clamp recordings comparing the electrophysiology of human neurons in BPI to BP (new Figure S7). No significant differences in the kinetics of evoked APs along with voltage-gated sodium and potassium currents analysed were found. The suboptimal number of calcium spikes seen in NEUMO and FluoroBrite media may be related to their formulations driving higher calcium store release and consequently suppressing neuronal activity by activating calcium-gated potassium currents and increasing action potential thresholds (Segal et al., 2018). This is a reasonable hypothesis. However, we could not obtain convincing evidence to substantiate this claim. Most importantly, as explained above, we are confident that BPI supports optimal electrophysiological activity and therefore is unlikely to significantly dysregulate calcium store release. We have now commented on this in the discussion (page 13, see lines 435-438 of the revised manuscript).

Minor Issues

1. It is not clear when the lentivirus is infected/transfected with neurons and when the experiments were performed after infection/transfection.

We apologize for this and thank the referee for this suggestion. We have now updated our 'lenti-viral vectors' methods section with these extra details (page 18, see lines 603-605 in manuscript). "Lenti-viral transfections in neuronal cultures were performed at least one week prior to experimentations. If expression levels were low, neurons were left in culture for an extra 3-5 days before further experimentations commenced." All lentiviral vectors used were titrated by quantitative real-time PCR to ensure sufficient MOI (multiplicity of infection) and transduction of the cells.

Reviewer #2 (Remarks to the Author):

In this paper the authors describe the development of a new neuronal culture media designed for optimal live-cell imaging of neurons in vitro. This media is an adapted formulation of the BrainPhys media first reported by Bardy et al. in 2015. In this paper, the authors adjust the formulation of this medium to reduce phototoxicity and autofluorescence, while still supporting electrophysiological activity of the cultured neurons. As the use of iPSC-derived neuronal cultures and organoids are becoming widely adopted, the development of media that support physiological neuronal activity and allow live-cell (imaging) studies – such as the BPI media described here - is highly relevant and will be of broad interest to the scientific community and the Nature Communications readership.

In general, the paper is well written, and the results support the claims. There are a few sections where the paper could be improved or the data could be strengthened, mostly through improving the analysis or data presentation. There are several sections of the methods that need to have significantly more information to allow others to reproduce the experiments.

We thank the reviewer for the positive and encouraging comments and insightful suggestions to improve the manuscript.

Major comments

2.1. In Figure 1B, it is impossible to judge the values for the lower autofluorescence media. Please make separate curves for that. The data can all be relative to PBS for comparison but the low intensity autofluorescence needs to be visible for the reader to judge the significance of the differences.

We appreciate this feedback and also agree with the comments. We have addressed this issue by presenting the original absorbance data in a heat map format. This can be found in new Figure 1B.

2.2. In Figure 1D-F, the autofluorescence of the media is significantly higher for the BP media relative to the BPI. However, in Figure 1G and 1I the values for the different media are surprisingly similar. This discrepancy needs to be explained.

This is a valid point. Thank you for pointing this out. To clarify this result, we now have added further experiments (new Figure 1C) and signal-to-background analysis (new Figure 2B), chose images more representative of the average quantification (new Figure 2A and E), provided an explanation for the discrepancies pointed out by the reviewer in the results and discussion sections (page 5, lines 147-153; page 12 - lines 423-424) and added more details in our methodology (pages 18-20, lines 627-684 and page 17, lines 570-587).

In Figure 1D-F (now Figure 1D-F) of the original manuscript, there are significant differences in RFU values of autofluorescence across the blue, green and red channels between BPI and BP conditions. The differences between BP and BPI in signal-to-background ratios in the blue and green channels shown in the graphs of original figure 1G and I (new Figures 2A,C, and 2E-F) are also significant. However, the reviewer is correct that the differences are less prominent in Fig 1G and I (new Figures 2A,C, and 2E-F) and not significant in the red channel for Fig 1G (new Figures 2E-F). The discrepancies are essentially due to differences in methodology and scales used for illustration in Figure 1 D-F. In Figure 1D-F (now Figure 1D-F), the media autofluorescence was measured using a BMG Labtech FLUOstar Omega which uses a light source at a 100 Hz flash rate and 167 ms exposure time, subjecting each well to 1mW of light power. This contrasts to imaging neuronal cultures in Figure 1G and I (new Figure 2A,C, and 2E-F), where light power intensities were minimized and mid-to-high NA lenses were used to reduce photobleaching. Images were acquired in Figure 1I (new Figures 2A,C) with a 0.8 NA Olympus 40x water immersion lens (Semi apochromat, LUMPLFLN40XW) and 200ms exposure time. GFP was excited using a PE300 (white) LED system set to 1%, which is equivalent to ~0.1 mW (~10 times less than in Figures 1D-F and now Figures 1D-F). In figure 1G (new Figure 2E-

F), a 0.45 NA Nikon 20x air immersion lens (Ph1 S Plan Fluor ELWD ADM) with a 25ms (NeuO), 204 ms (Hoechst) or 500 ms (B-III tubulin) exposure time was used. All fluorophores in Figure 1G (new Figure 2E-F) were excited using a Lumencor Sola Light (SE 5-LCR-QB) built into the ImageXpress Micro 4 High Content Screening System, which was set to 100 “lumencor intensity” units in MetaXpress (v6.2.2.) software. Even though methodologies likely contributed the most to this discrepancy, it is important to note that RFU values for BPI and BP in the red channel (Figure 1F, new Figure 1F) are also much less than for blue and green. This also contributes to the similarities seen between BPI and BP for B-III tubulin signal-to-background ratios (Figure 1G, new Figure 2E-F).

To more accurately represent the size of the difference between the autofluorescence of the different media we have collected new autofluorescence data measuring emission spectra intensities from test media across the visible spectrum following various excitation wavelengths (375, 405, 488 and 532nm). The results are presented in heat map format (new Figure 1C) and more clearly display changes in RFU values across all channels using a comparable fluorescence intensity scale. It can be seen following light excitation at 532 nm that autofluorescence is similar to PBS for all test media across the visible light spectrum (new Figure 1C), which aligns with the similarity between conditions seen in Figure 1G (new Figure 2E-F). Our results and discussion have since been updated to explain these findings (see pages 4-5, lines 127-136 and 147-153; page 12 - lines 423-424). Representative images have also since been adjusted to more appropriately represent signal-to-background ratio quantifications (see comment 2.3; new Figure 2A,E), and additional signal-to-background analysis added (see comment 2.4; new Figure 2B). Detailed ‘*Autofluorescence Measurements of Culture Media*’ (page 17, lines 570-587) and ‘*Imaging Acquisition*’ (pages 18-20, lines 627-684) methods have been included to further explain these different methodologies, as requested in minor comments 6(a-d), 7 and 9.

2.3. The quantification of fluorescence background in Figure 1G doesn't seem to reflect the images that are shown. Are the intensity display settings the same for all the panels in Figure 1G? If not explain why and how they were adjusted.

Thank you for the feedback. In Figure 1G of the original manuscript, window/level contrast settings of example images for blue, green and red fluorescence channels were kept constant in both BP and BPI, respectively. In our resubmission, we have readjusted raw images with the following maximum/minimum intensity counts: 8500/0 (Hoechst), 3600/0 (NeuO), 12000/1000 (B-III tubulin) (new Figure 2E). These image adjustments now more accurately reflect the background and signal intensity quantifications shown in new Figure 2F of the updated manuscript. All adjustments made to representative images have been included in new figure legends 2D,E,S1F, and S3, along with lens NA, immersion liquid and magnification details.

Furthermore, example GFP (green fluorescence) images have now been updated by applying a LUT that displays changes in the background and signal intensities across a colour gradient (new Figure 2A and S2A). Cropped images at the soma, neurites and background regions of neurons in either BP or BPI were also added to highlight changes in signal and background intensities between media at these regions. These examples reflect the data quantified in new Figures 2B and 2C. Changes in GFP intensities at these regions have been addressed in the results (page 5, lines 160-167) and corresponding figure legends have been updated.

2.4. It is unclear if the background intensity is subtracted before measuring the signal. It is important to do that, or it will severely impact the S/B ratios for the different wavelengths. If the signal intensity was background corrected, then the differences in signal to background ratio seem to come primarily from an increase in signal, rather than a reduction of the background, in contrast to what is stated on page 5, line 140. If it was not the data needs to be recalculated.

We appreciate this feedback and apologize for not making this clearer earlier. To better address this point we have conducted further analysis of all signal-to-background ratios which were shown in Figures 1G,I and S1 of the original manuscript. We have recalculated the signal-to-background ratios for each field-of-view following Weber's Contrast formula: $((\text{signal intensity} - \text{background intensity})/\text{background intensity})$. This new data can be found in new Figures 2B, C, F, and S1C-E of the updated manuscript. Corresponding figure legends and ‘*fluorescent signal-to-background ratio*’ methods have been updated to include these calculation details (see page 20, lines 708-710 and page 21, lines 727-728).

In addition, since there is a larger amount of GFP in the cell body compared to neurites, we have now separately analysed the mean intensity at the soma, dendrites and background regions of human neurons expressing GFP (new Figure 2B and S2B). Data show that BPI improved the signal-to-background ratio relative to BP at isolated neurite and soma regions, and also reduced background intensity. This has now been addressed in the results (see page 5, lines 161-166). Updated methods for this GFP image analysis have also been included (page 21, see lines 732-739).

2.5. Many details are missing in the methods section. See minor comments below for details. The way the methods are now the imaging and image analysis could not be reproduced.

We thank the reviewer for this useful feedback. Methods on image analysis and imaging have been updated under the following sub-headings with details requested in minor comments: 'Quantification of Cell Viability with Imaging for Phototoxicity Experiments' (page 20, see lines 686-700), 'Image Acquisition' (page 19-20, see lines 626-684) and 'Fluorescent Signal-to-Background Ratios' (page 20-21, see lines 702-730). See specific details also below.

2.6. Were antibodies validated?

Yes, the antibodies used in this study have been validated in the SCT lab against appropriate positive and negative controls for immunocytochemistry. The three antibodies in question are well cited for our application and cell types (human and rat neurons). A statement about antibody validation is also included in the Reporting Summary with more information. Thank you for your question.

2.7. Fig 1I: the results for ACSF are not mentioned in the results section, they should be discussed.

Thank you for picking this up. ACSF results have now been discussed for Figure 1I (new Figure 2C) in the results section of our updated manuscript (see page 5, lines 161-166) :

'... The human neurons were imaged live in standard BrainPhys or BPI with identical imaging parameters. BrainPhys Imaging was found to significantly increase signal-to-background ratios at neurite and soma regions, and significantly reduce mean background intensities (Figure 2A-C). When combining analysis at soma and neurite regions, a similar increase in signal-to-background ratios was observed. Re-perfusing the cells with ACSF, after testing them subsequently in BPI and BP, recovered signal-to-background ratios at the background, soma and/or neurite regions to levels identical to BPI (Figure 2A-C). This confirmed that the effect was not simply due to possible photobleaching.'

2.8. In Figure 2, a quantification of live neurons pre-light should be shown in B, D and F to establish a baseline. The results could be presented as a change in cell number.

We appreciate this excellent feedback. We have now normalized these post-exposure results in Figure 2B, D and F of the original manuscript to the mean number of neurons pre-light exposure in each medium (now new Figure 3D). Results show the percentage loss in cells following exposure to violet, blue and red light. Corresponding figure legends have been updated with these changes.

2.9. Experiments showing photobleaching of the different dyes in the different media would give an idea if dye photobleaching is contributing to phototoxicity.

We thank the referee for this suggestion. Establishing a robust causal relationship between the photobleaching characteristics of the fluorophores, the concentration of reactive oxygen species and phototoxicity of the system investigated here would be very interesting. However, disentangling cause and effect in this system is relatively challenging since reactive oxygen species (ROS) are both created by most fluorophores AND cause them to photobleach. Hence, a higher photobleaching rate may indicate that a particular dye contributes more ROS to the system OR that more ROS were present in that system to start with, which causes a faster photo-bleaching. Deciding which of the two is the case would be required to understand the role of the fluorophores in the creation of phototoxic molecular species as suggested by the referee. However, this is a particularly challenging task requiring many

carefully designed and labour intensive experiments, which we believe are beyond the scope of this study and our current experimental capacity.

2.10. Experiments showing endpoint intensity of a ROS probe or mitochondrial morphology would help demonstrate phototoxicity with the different media. This may pick up more subtle phototoxic effects that do not show up as cell death for example.

We thank the referee for this excellent suggestion. To address these comments, we have performed additional experiments, which are illustrated in Figure 3 and showed that LDH increased over time after exposure to relatively low ambient light of BP but not BPI (new Figures 3E-F), ROS increases in BP but not BPI (H_2O_2) (new Figures 3G, J), and light-induced impairment of mitochondrial potential in BP but not BPI (new Figures 3K). The results were described (see page 6-7, lines 200-219), the methods were added (see pages 22-23, lines 750-809) and Figure legends were updated.

2.11. There is a statement in the methods section that contrast and exposure settings were adjusted between timelapse imaging sequences when necessary. Does that mean the light exposure of the samples differs for different media tested? If this is the case this must be carefully incorporated into the results to make sure differences in intensity are not playing a role in the observed phenotypes.

Thank you for this comment and we apologize for not making this clearer. Contrast and exposure settings were adjusted between time-lapse sequences to ensure they were the same in different media tested. Exposure times were fixed at 200ms across all calcium-imaging protocols. This has now been made clearer in the '*Calcium Imaging*' method section (see page 24, lines 828-829 and 832-833 in manuscript).

Minor comments

1. Avoid using terms like better and worse be specific. For example, on lines 129 and 139 just say autofluorescence is lower, not better. Similarly, the Fig. 1 title should be signal-to-background ratio is higher... not improved.

Thank you for this feedback. We have now corrected this throughout the updated manuscript.

2. All cell images should be in greyscale unless they are part of a colour overlay.

We appreciate this suggestion. Greyscale images have now been added to the main and the supplementary Figures of the updated manuscript (new Figures 2D, E, S1F, and S3A).

3. In Figure 1G and 1I, use a gamma factor to show neuronal protrusions and autofluorescence without saturating the cell body. Put these image display properties in the figure caption so readers know it is a non-linear display, but this will avoid the saturated cell bodies in the images.

We thank the referee for this comment. We found that adjusting example images with a *gamma factor* to show neuronal protrusions and autofluorescence did not represent very well the differences in signal and background intensities quantified between BP and BPI. Instead, to address the reviewer's suggestion, we have adjusted the images in previous Figure 1I and G (new Figure 2A and E) with maximum and minimum intensity counts or had a LUT applied to visualize GFP intensities at soma, neurite and background regions. Further details are outlined in comment 2.3 above. All adjustments to raw images have been included in updated figure legends: 2E, S1F and S3A.

Images in Figure 1G and I were taken using a 16-bit camera, where the maximum number of counts for every pixel is 65,536 ($= 2^{16}$). The highest counts found for all GFP images in previous Figure 1I were ~10,000 counts, while most image features (including the nucleus) are in the 200 - 2000 count region (new Figure S2B). Ultimately, no physical saturation occurred in any of the acquired images and any apparent saturation was likely related to the dynamic range shown in representative images of the original manuscript which has since been updated (see comment 2.3). Camera specifications and bit-depth (page 18, lines 630) used for imaging acquisition have been included in the updated manuscript, along with additional imaging specifications requested in minor comments 6(a-d) and 7.

4. Plots in Figure 1G should go to 0. Add a second scale break on the y-axis if necessary.

We thank the referee for this comment. The y-axes of plots in Figure 1G from the original manuscript now go to zero (new Figure 2F).

5. In Figure 1B, the combination of colors and thickness of the lines, makes some curves barely visible and the plots hard to read.

Thank you for the feedback. For more clarity, we have now converted the data in Figure 1B to a heat map (see new Figure 1B of the updated manuscript).

6. In depth details of the following are missing:

a. Imaging parameters for the ImageXpress Micro 4 HCS

We thank the referee for this suggestion. Requested details have been added to '*Image Acquisition*' (see page 18-19, lines 632-640, 647-653, 659-665 of the manuscript).

b. Imaging parameters for the SP8

These requested details have been updated under '*Image Acquisition*' (see page 18-19, lines 641-645, 653-657, and 669-672 of the manuscript).

c. Image analysis settings for the custom module designed for neuron quantification in MetaXpress 6.2.2

We appreciate this feedback. These requested details have been updated under '*Quantification of Cell Viability with Imaging for Phototoxicity Experiments*' (page 20, see lines 686-700).

d. Imaging parameter for the Olympus IX73 – e.g. what are the properties of the 40x lens? Lens type, NA, camera exposure time, light source?

We appreciate this feedback. These details have been updated and can be found in our methods section under '*Image Acquisition*' (see page 20, lines 674-682 in manuscript).

- '*Fluorescent images of live hPSC-derived neurons transfected with green-fluorescent-protein (GFP) lentivector were imaged at 200ms exposure time using a PCO.Panda 4.2 (16-bit) digital camera and Olympus BX51 upright microscope with 40x water immersion lens (LUMPLFLN40XW, 0.8NA, Semiapochromat). A cool-LED pE300 illumination unit was set to blue (460 nm) at 1% power (0.1 mW) and light passed through a single band FITC/Cy2 filter (Chroma, 49002) with a 470 nm excitation (± 40 nm) and 525 nm emission (± 50 nm) wavelength. Single neurons were perfused with BrainPhys™ basal media bubbled with a mixture of CO₂ (5%) and O₂ (95%) and maintained at room temperature. Perfusates were then switched to BrainPhys™ Imaging and then ACSF.'*

7. In the methods section, please provide the properties of the objectives used (not only magnification, but also NA and color correction (Neofluor, apochromat, plan-apochromat?))

This has been corrected for and updated throughout our methods section under '*image acquisition*' (pages 18-20, see lines 626 to 684 in manuscript). Key details are listed below:

- *An Olympus BX51 upright microscope was used with an Olympus 10x (UMPLFN10XW, 0.3 NA, Semi-Plan Apochromat) or 40x water immersion lens (LUMPLFLN, 0.8 NA, Semi-Plan Apochromat). See Figures 2A-C, 5.*
- *Phase contrast images were taken using an Olympus CKX53 microscope with a 10x dry objective (CACHN10XIPC, NA 0.25). See Figures 3A-D, S4B, S5 and 7A.*

- *Leica SP8 confocal microscope was used with a 10x dry objective (HC PL APO CS2, NA 0.4) or 63x oil objective (HC PL APO CS2; NA 1.4). See Figures 2D and 7A.*
- *ImageXpress Micro 4 High Content Screening System was used with a 20x objective (Ph1 S Plan Fluor ELWD ADM; NA 0.45). See Figures 2E-F, 7B and S1C-F.*

8. In Figures 3-5, why wasn't BP media included as a control?

We thank the referee for this comment. Previously, we have demonstrated that the electrophysiology of human neurons in BP and ACSF is comparable (Bardy et al. 2015; PMID: 25870293). ACSF is usually the gold standard for acute electrophysiological experiments and therefore we originally decided to essentially compare BPI to ACSF. Given that figures 1-2 of the original manuscript showcase that BP promoted phototoxicity, autofluorescence across the visible light spectrum and reduced signal-to-background ratios, it was not considered necessary to test BP with functional imaging applications.

However, to address this comment, we have performed further experiments and added the following:

1. Patch-clamp recordings from human neurons in BPI compared to BP were conducted and the kinetics of evoked APs along with sodium and potassium currents analysed (new Figure S7).
2. Examples of synaptic activity, evoked APs and I-V traces shown from the same patch-clamped neuron in ACSF, BP and BPI (new Figure S7).
3. Multi-electrode (MEA) array recordings of the network activity from human neuronal cultures in BP and BPI (new Figure 4Q-V).

These new results highlight no significant differences between BP and BPI in maintaining the physiological activity of neuronal cultures at a single cell and network level. Figure 4W-X of the updated manuscript (previously figures 3Q-R) also show that BPI maintained spontaneous firing rates comparable to BP during long term culture. The results have been addressed in the manuscript (see page 7, lines 234-235, and page 8, lines 255-258).

9. More information on the autofluorescence media measurements with the BMG Labtech is needed. What is the bandwidth of the filters, what light source was used, intensity of light, exposure time, what detector, etc?

We apologise that these details were not listed in the manuscript. We have now updated our 'autofluorescence' methods with the information requested (page 17, see lines 578-587).

'Equal volumes (300 μL) of control and test media were pipetted into a CellCarrier-96 black plate with an optically clear bottom (Perkin Elmer). A total of eight replicate wells per medium were tested with a BMG Labtech FLUOstar Omega using an excitation/emission of 355/460 (blue), 485/520 (green), 544/590 (red) and 584/620 nm (far-red). FLUOstar uses a high energy xenon flash lamp light source with a 100 Hz flash rate and 167 ms exposure time. The energy per flash emitted is 67 mJ and subjects each well to 1mW of light power. Emission wavelengths were captured using a BMG Labtech photomultiplier tube detector (PMT) at a gain of 1000. Blue, green, red and far-red filters utilized the respective bandwidths for their excitation/emission wavelengths: 20/20, 12/25, 20/20 and 10/20 nm. Acquired autofluorescence data was blank subtracted. For normalization, the mean fluorescence intensity in PBS was subtracted from the other media.'

10. Secondary antibody catalogue numbers, lot numbers and concentrations are missing.

We apologise that these details were not listed in the manuscript. Concentrations for each antibody have been added throughout the 'Immunofluorescent Staining Protocol' in the methods section (page 18, see lines 607-617 of the manuscript). Lot number and catalogue numbers have been updated in the supplementary resource table.

11. Catalog and lot numbers (if appropriate) should be added for all reagents and tools mentioned in the Methods section.

We thank the referee for this suggestion. All catalogue and lot numbers (where applicable) for reagents, antibodies, biological samples, and additional resources can be found in the supplementary key resource table with corresponding source details.

12. This article should be cited. "Assessing phototoxicity in live fluorescence imaging, P Philippe Laissue, Rana A Alghamdi, Pavel Tomancak, Emmanuel G Reynaud & Hari Shroff, Nature Methods volume 14, pages 657–661(2017).

Thank you for this paper, it is very relevant and we have now cited this within the introduction (see page 3, line 68).

13. ImageJ and/or FIJI papers and plugins should be properly referenced.

We appreciate the suggestion, and have added the following reference into our manuscript where used for image analysis and calcium imaging (see page 20, line 705, page 21, lines 714,725,734; page 24, line 841):

In text: Fiji (ImageJ) software (Schindelin et al., 2012).

Ref: *Schindelin, J.; Arganda-Carreras, I. & Frise, E. et al. (2012), "Fiji: an open-source platform for biological-image analysis", Nature methods 9(7): 676-682, PMID 22743772, doi:10.1038/nmeth.2019 (on Google Scholar).*

14. Were any core facilities used for this work that should be acknowledged?

Thank you for this correction. This is now updated as follow (see page 27, lines 952-956):

Flow cytometry analysis and cell sorting were performed at the South Australian Health Medical Research Institute (SAHMRI) in the ACRF Cellular Imaging and Cytometry Core Facility. The Facility is generously supported by the Detmold Hoopman Group, Australian Cancer Research Foundation and Australian Government through the Zero Childhood Cancer Program.

Reviewed by Claire M. Brown, McGill University with assistance from Michiel Krols, Post Doctoral Fellow, McGill University as a review training exercise.

We really appreciate the reviewer's feedback. We have done our best to address them all within the timeframe allocated and we believe they have significantly improved our manuscript.

Reviewer #3 (Remarks to the Author):

Significant advances have been recently made in two major in vitro technologies: the synthetic control of cell fate, and the ability to optically measure and manipulate activity in these cells. However, the artificial cellular environment or media that is used to maintain and functionally monitor these cells - particularly neurons - has not seen a parallel advance. This has resulted in unwanted side-effects such as phototoxicity in long-term cell culture experiments, inability to faithfully capture cell-physiological changes, and a concomitant reduction in cell viability. In the study by Zablocki et al., the authors improve upon their previous formulation (BrightPhys or BP) to generate a new cellular media (BrightPhys Imaging or BPI) which they claim is better for measuring and manipulating in vitro neuronal activity, especially when studying human stem cells in culture.

The authors perform a series of experiments to test fluorescence performance in fixed tissue, and then test their media in a wide range of in vitro applications, including Ca²⁺ imaging and optogenetics. Based on their data, they make the claim that BPI is a superior media formulation compared to other commercially available options. While the updated formulation (BPI) is indeed better in terms of signal-to-background in some applications and ranges of excitation/emission, it is really not demonstrated that this new formulation is a significant advance over their previous one (BP), especially considering that the range and size of overall benefit do not seem that much higher. In addition, the fact that they can generally capture and manipulate physiology in human neuronal cultures has already been previously shown by the same authors.

The major general issues with the paper are:

1) The whole paper is only a very incremental step forward compared to their previous study – and not a significant conceptual advance that would deserve publication in a high impact journal; 2) The study is mostly descriptive, lacking mechanistic data as to why BPI seems to perform better in some aspects compared to its predecessor and other commercially available media, and 3) The paper is purely methodological and as a method is not particularly novel. Since the corresponding author has already published a paper on the BP formulation previously and has covered very similar ground in that paper, the novelty of this report is significantly reduced.

We thank the reviewer for the feedback and suggestions, which we have carefully addressed in the revised version. The constructive comments significantly improved the manuscript and we are most grateful for them. We strongly believe that this work will be very valuable to the research community in particular in the field of neuroscience as also clearly stated by reviewers #1 and #2. We hope that reviewer #3 will be pleased with the revised version and find it suitable for publication.

Specific weaknesses

3.1. The signal-to-background analysis, on which one of the main claims is based, is relatively simplistic. For example:

a. How does the signal-to-background ratio scale with the actual level of GFP? In Fig. 1. I, the authors have quantified mean GFP intensity in the cell body and dendrites together. Since there is a larger amount of GFP in the cell body, this region seems saturated. This potentially obscures the true signal-to-background ratio. The authors must compare quantities based on a reasonable dynamic range, and avoid saturating pixels.

We thank the reviewer for this excellent feedback. We have now clarified this issue with new analysis. We agree with the referee that since there is a larger amount of GFP in the cell body compared to neurites, this could have hidden the true signal-to-background ratio calculated in Figure 1I of the original manuscript. To address this, we have now analyzed the mean intensities within the soma, dendrites and background from images of human neurons expressing GFP taken in ACSF, BP, and BPI. We have included the new analyses in the updated manuscript as follow:

- Signal-to-background ratios at neurites and/or soma regions (new Figure 2B-C).
- Mean background intensities (new Figure 2B).
- Mean GFP intensities at soma, neurite and background regions against the cumulative % of pixels (new Figure 2B) and number of pixels (new Figure S2B).

This new data shows an improved signal-to-background ratio at neurites and/or soma regions in BPI, but also a reduced mean background intensity, relative to BP. To maintain a comparable and reasonable dynamic range, mean GFP intensities at these regions were also plotted against the cumulative % and the number of pixels. This new data suggests that improvements in signal-to-background ratios seen in Figure 1I of the original manuscript are due to a combined effect: increased soma and neurite intensities, and a reduced background intensity. 'Fluorescent Signal-to-Background Ratio' methods have been updated (see page 20, lines 703-712) and additional methods added under 'Image Analysis Per Pixel' (page 20, see lines 732-739 in manuscript). These results have now been addressed in our updated manuscript (see page 5, lines 161-166).

GFP Images in Figure 1I (new Figure 2A) were taken using a 16-bit camera, where the maximum counts for every pixel is 65,536 (= 2^{16}). The highest counts found for all GFP images were ~10,000 counts, while most image features (including the nucleus) were in the 200 - 2000 count region (see new Figure S2B). Ultimately, no physical saturation occurred in any of the acquired images. We have since updated these example images by applying an LUT which highlights the mean GFP intensity at the soma, neurites and background regions across a colour gradient (see new Figure 2A). Camera specifications and bit-depth used for imaging acquisition has been included in the updated manuscript along with further imaging details (see pages 20, lines 674-679).

b. How does the signal-to-background ratio compare across media in Ca²⁺ imaging experiments?

Fluo4-AM has a peak excitation/emission at 494/506 nm. We have collected new autofluorescence data measuring emission spectra intensities from test media across the visible spectrum following various excitation wavelengths, including 488 nm (new Figure 1C). The results are presented in heat map format (see new Figure 1C and page 4 -5,

lines 127-136 of the manuscript). Light excitation at 488 nm yields a similar level of autofluorescence at 506nm Emission in PBS for BPI, NEUMO and FluoroBrite test media (new Figure 1C). The absorbances of the imaging media are also very similar to PBS between 494nm and 506nm (Figure 1B). Therefore, we do not expect major differences in the fluorescence detection for the imaging media tested for the Ca²⁺ experiments. However, different concentrations of calcium and other neuroactive components in the media can affect the cellular levels of calcium and in turn the Fluo-4 signal intensity. Such electrophysiological differences impair the calcium spike activity (Figure 5E, F, I, J) but not the slow rising calcium waves (Figure 5G, H, K, L). This was also confirmed with patch-clamping (Figure 4, S10). To further address the reviewers question, we have analysed the time-lapse image sequences of human neurons incubated with 1uM Fluo4-AM and imaged under 10x magnification at 5Hz. Each field-of-view was recorded in all 3 perfusates (NEUMO, BPI and FluoroBrite) with a 200ms exposure time. The mean intensity at the cell bodies (signal) and background regions in each field-of-view was calculated across the entire 4-minute image sequence. No significant differences were found between signal-to-background ratios computed for BPI, Fluorobrite and NEUMO. In addition, the calcium event analysis was normalised to the resting fluorescent level for each cell (see 'Calcium Imaging' methods). In this case, we are therefore confident that the differences in calcium spikes activity between imaging media is caused by electrophysiological impairments rather than discrepancies in fluorescence detection.

3.2. The claim that BPI does not result in major phototoxic cell death compared to other media is a very significant claim. However, other than a very surface level exploration of photo-toxicity, there is a lack of explanation as to why BPI does not lead to a significant loss in cell density compared to other media.

a. In Fig.2, the authors compare the effect of light exposure on the viability of primary rat cortical neurons in different media. However, this is a very surface level analysis. The authors must show further details. For example, how temporally dynamic is the loss of cells? Since light excitation/exposure conditions vary widely across experiments, how does the level of exposure scale with loss of cells?

b. What is the mechanism of protection that BPI confers on these cultures? Are there differences in gene/mRNA/protein expression following light exposure across media that provide an explanatory mechanism? Did the authors measure reactive oxygen species and other cytotoxic components in the media post-exposure? At what level does the BPI prevent major cell death?

We thank the reviewer for these excellent suggestions. To further substantiate the claim of phototoxicity, we have performed additional experiments which are now in Figure 3E-K. We have now looked at the cytotoxic effect of light on the health of hiPSC-derived cortical and midbrain neurons comparing BP and BPI. We measured reactive oxygen species concentrations and mitochondrial potentials and levels of lactate dehydrogenase in supernatant across multiple light intensities, now also including the effect of ambient light in the tissue culture hood. The new results are described (see page 6-7, lines 200-219), and Figure 3E-K associated legends and methods have also been updated (see pages 22-23, lines 750-809).

c. In Fig. 2B-F, the authors compare cell counts post-exposure. Instead, the authors should compare percentage loss in cells pre- and post-exposure.

We appreciate this excellent feedback. We have now normalized these results in Figure 2B, D and F of the original manuscript to the mean number of neurons pre-light exposure in each medium (see new Figure 3D). Results show the percentage loss in cells following exposure to violet, blue and red light. This new analysis has been updated for corresponding legends.

3.3. Comparison of BPI to BP physiology experiments – did the authors compare BP to BPI in the electrophysiology, Ca²⁺ imaging, and optogenetics experiments? If BP is comparable to BPI in terms of physiology, then it is a point against the significance of the novelty of BPI compared to BP.

BPI is reformulated to maintain the physiological activity of neuronal cultures, reduce phototoxicity and improve fluorescence signals throughout the entire light spectrum. In Figures 1-2 of the original manuscript (now Figures 1-3), we show the BP standard promotes phototoxicity and reduces fluorescence signals. We have since added further data to strengthen these claims as requested (see comments to 3.1a and 3.2a,b). Although it is possible to measure

calcium imaging and perform optogenetics in BP or any other medium, in view of the new results presented in Figure 1-3, live imaging of neurons in standard BP will be suboptimal and potentially harmful for the neurons depending on the light intensity levels and duration of exposure. However, we also showed that the benefit of using standard BP to support optimal electrophysiological properties are maintained in BPI, which was an important objective in the design of BPI. We have now performed further experiments and added new data in the revised manuscript to further substantiate this claim (new Figure 4Q-V, new Supp Figure 7, see also Figure 4W). These new results have been addressed in the manuscript (see page 7, lines 234-235, and page 8, lines 255-258). Corresponding figure legends have been updated.

Figures 5-6 (now Figures 6-7) show that media such as BrightCell NEUMO and FluoroBrite did not support the physiological activity of neuronal cultures in imaging applications. Given that NEUMO and FluoroBrite are respectively reformulated from Neurobasal and DMEMF/12, our data aligns with our previous paper (Bardy et al. 2015; PMID: 25870293) which shows the negative impact of DMEMF/12 and Neurobasal on the electrophysiological properties of neuronal cultures.

The novelty of BPI is based on the improvement of the quality of a wide range of fluorescence imaging applications with live neurons *in vitro* compared to BP while outperforming neurophysiological support compared to alternative imaging media. As the use of iPSC-derived neuronal cultures and organoids are becoming widely adopted, the development of a new medium which supports both physiological neuronal activity and live-cell (imaging) studies is novel and will be of significant interest to the growing scientific community using imaging tools and iPSC-derived neuronal cultures.

3.4. With this manuscript, the authors make only a small incremental jump from their previous publication. Being a person who images in cultures is an excellent neural resource; however, several things reduce my enthusiasm for this manuscript. I think the authors do an outstanding job in Figure 1, demonstrating the difference between BP and BPI in terms of signal to noise. In Figure 2, I believe the authors continue to provide convincing data showing the difference between BP and BPI to Photo-toxicity via violet, blue, and red light. However, the major issues for this manuscript start with figure 3. It would have been beneficial for the authors to continue to make a direct comparison between BP, BPI, ASCF throughout Figure 3 and others.

We thank the reviewer for the positive and encouraging feedback on our work. It is much appreciated. We also appreciate the critical feedback for Figure 3 and we have taken the opportunity to improve it by performing new experiments and add a side-by-side comparison between BP and BPI with another measurement: synchronicity of neural network activity in human neurons (new Figure 4Q-V). We have also conducted more patch clamping experiments to compare the effect of ACSF, BP and BPI on human neurophysiological properties (see new supplementary Figure 7), as suggested. These new results confirmed that there are no significant differences between ACSF, BP and BPI in maintaining the physiological activity of neuronal cultures at a single cell and network level, which was an important objective when we designed BPI. These new results have been addressed in the manuscript (see page 7, lines 234-235, and page 8, lines 255-258). Corresponding figure legends have been updated.

3.5. In Figure 4, I was surprised the authors used Fluo-4AM to test the Ca²⁺ signaling in the human neuronal cultures. For example, the authors could be missing some of the Ca²⁺ signals due to the harmful effects of Fluo-4AM on neuronal properties via disruption of the Na, K ATPase. I recommend the authors review the downside using Fluo-4AM and such indicators as in Smith et al., 2018 PMID: 29382785. It would have been beneficial for the authors to use viral GCaMP to assessed Ca²⁺ signaling in human neurons. The authors also failed to include BP in the comparison between BPI and ASCF. This was another lost opportunity for the authors to demonstrate why BPI is superior to BP.

We thank the referee for raising this important point. We have read the paper referenced (Smith et al., 2018; PMID: 29382785) with great interest. In our experiments, we specifically used a very low dose of Fluo-4 AM (1 μ M) with a 20-minute loading time, which does not have any significant harmful effect or influence on neuronal properties via disruption of the Na, K ATPase and also cell viability. For comparison, in Figure 1b of Smith et al., the intracellular concentration of Fluo-4 AM is <0.3 mM after loading a 1 μ M extracellular concentration. In Figure 1c, the ouabain

(1mM)-sensitive 86Rb+ uptake with pre-incubation of Fluo-4 AM <0.3 mM intracellular concentration was found to not be significant when compared to without incubation. In Figure 2 of Smith et al., the Na,K-ATPase activity and 86Rb+ uptake was measured in mouse neurons (2f) after 2 and 5µM Fluo-4AM was preloaded for 30 minutes. At 2µM concentrations of Fluo-4 AM, no significant differences were found from the control for Na,K-ATPase activity and 86Rb+ uptake. Furthermore, for human astrocytes (2c) the 86Rb+ uptake after 2µM Fluo-4 AM loading for 30 minutes was not significant from the control, and *p<0.05 for Na,K-ATPase activity. In both cases, 2 µM Fluo-4 AM is double the concentrations with loading time 30% longer than those used in our experiments. Lastly, Figure 5a showcases that 1µM concentrations of Fluo-4 AM had no significant differences compared to controls on cell viability (%) when loaded over 30 minutes. Our protocol utilised a 20-minute loading time. Overall, Smith et al. paper suggests that no significant changes in cell viability (%), or disruption of the Na, K ATPase should be found in our neuronal cultures as we utilised a 1 µM Fluo-4 AM concentration with a loading time of 20 minutes. This is an important point and we have now made it clear in the methods '*Calcium Imaging*' methods so our incubation time and final Fluo4-AM (1µM) concentration used is clearer (see page 24, lines 821-824). This paper (Smith et al., 2018) has now been cited in our updated manuscript (see page 24, lines 827-828).

3.6. In Figure 5, the authors missed another opportunity to compare BP, along with BPI and ACSF.

Following the reviewer suggestions, we have conducted additional experiments to compare BP along with BPI and ACSF (see new supplementary Figure 7, Figure 4, and Bardy et al. 2015 for comparison between BP and ACSF, PMID: 25870293). Figures 1-3 demonstrate that BP is suboptimal for imaging or optical experiments. Therefore we focused our functional imaging and optogenetic experiments comparing BPI with ACSF and other specialised imaging media.

Reviewers' Comments:

Reviewer #1:

Remarks to the Author:

To the questions I came up with, I think the authors have provided an accurate and satisfactory answers. I don't feel the need to ask any further questions.

Reviewer #2:

Remarks to the Author:

The authors did an excellent job addressing my concerns for the manuscript. I appreciate the significant efforts that went into adding details to the methods aspects of the paper. These will be important if others want to reproduce this work.

I have two comments that need to be addressed before publication:

- 1) The company names were added but the catalog numbers need to be added for all reagents.
- 2) It is essential to add the catalog numbers AND the lot numbers for all the antibodies that were used.

Reviewer #3:

Remarks to the Author:

The authors have addressed all issues raised in the previous review cycle and added a significant body of new experiments.